# Understanding the Complexity Gains of Reformulating Single-Task RL with a Curriculum

## Abstract

Reinforcement learning (RL) problems can be challenging without well-shaped rewards. Prior work on provably efficient RL methods generally proposes to address this issue with dedicated exploration strategies. However, another way to tackle this challenge is to reformulate it as a multi-task RL problem, where the task space contains not only the challenging task of interest but also easier tasks that implicitly function as a curriculum. Such a reformulation opens up the possibility of running existing multi-task RL methods as a more efficient alternative to solving a single challenging task from scratch. In this work, we provide a theoretical framework that reformulates a single-task RL problem as a multi-task RL problem defined by a curriculum. Under mild regularity conditions on the curriculum, we show that sequentially solving each task in the multi-task RL problem is more computationally efficient than solving the original single-task problem, without any explicit exploration bonuses or other exploration strategies. We also show that our theoretical insights can be translated into an effective practical learning algorithm that can accelerate curriculum learning on robotic goal-reaching tasks.

## 1 Introduction

Reinforcement learning (RL) provides an appealing and simple way to formulate control and decision-making problems in terms of reward functions that specify what an agent should do, and then automatically train policies to learn how to do it. However, in practice the specification of the reward function requires great care: if the reward function is well-shaped, then learning can be fast and effective, but if rewards are delayed, sparse, or can only be achieved after extensive explorations, RL problems can be exceptionally difficult (Kakade and Langford, 2002; Andrychowicz et al., 2017; Agarwal et al., 2019). This challenge is often overcome with either reward shaping (Ng et al., 1999; Andrychowicz et al., 2017; 2020; Gupta et al., 2022) or dedicated exploration methods (Tang et al., 2017; Stadie et al., 2015; Bellemare et al., 2016; Burda et al., 2018), but reward shaping can bias the solution away from optimal behavior, while even the best exploration methods, in general, may require covering the entire state space before discovering high-reward regions.

On the other hand, a number of recent works have proposed multi-task learning methods in RL that involve learning contextual policies that simultaneously represent solutions to an entire space of tasks, such as policies that reach any potential goal (Fu et al., 2018; Eysenbach et al., 2020b; Fujita et al., 2020; Zhai et al., 2022), policies conditioned on language commands (Nair et al., 2022), or even policies conditioned on the parameters of parametric reward functions (Kulkarni et al., 2016; Siriwardhana et al., 2019; Eysenbach et al., 2020a; Yu et al., 2020b). While such methods are often not motivated directly from the standpoint of handling challenging exploration scenarios, but rather directly aim to acquire policies that can perform all tasks in the task space, these multi-task formulations often present a more tractable learning problem than acquiring a solution to a single challenging task in the task space (e.g., the hardest goal, or the most complex language command). We pose the following question: can we construct a multi-task RL problem with contextual policies that is easier than solving a single-task RL problem from scratch? In this work, we answer this question affirmatively by analyzing the sample complexity of a class of curriculum learning methods.

To build the intuition for how reformulating a single-task problem into a multi-task problem enables efficient learning, consider the setting where the optimal state visitation distributions $d_\mu^{\pi_\omega^\star}, d_\mu^{\pi_{\omega'}^\star}$ of two different contexts $\omega, \omega'$ are "similar", and our goal is to learn the optimal policy $\pi_{\omega'}^\star$ w.r.t. $\omega'$. Suppose we have learned the optimal policy $\pi_\omega^\star$, we can facilitate learning $\pi_{\omega'}^\star$ by: (1) using $\pi_\omega^\star$ as

an initialization and (2) setting a new initial state distribution $\mu' = \beta d_\mu^{\pi_\omega^\star} + (1 - \beta_\mu)$ by mixing $d_\mu^{\pi_\omega^\star}$, the optimal state visitation distribution of $\pi_\omega^\star$, and $\mu$, the initial distribution of interest. Using $\pi_\omega^\star$ as initialization for learning $\pi_{\omega'}^\star$ facilitates the learning process as it guarantees the initialization is within some neighborhood of the optimality. Setting the new initial distribution $\mu' = \beta d_\mu^{\pi_\omega^\star} + (1 - \beta)\mu$ also facilitates the learning of $\pi_{\omega'}^\star$ as it reduces the density mismatch ratio between $\left\| d_\mu^{\pi_{\omega'}^\star} / \mu' \right\|_\infty$, a key quantity that influences sample complexity. The new distribution $\mu' = \beta d_\mu^{\pi_\omega^\star} + (1 - \beta)\mu$ for learning $\pi_{\omega'}^\star$ could be understood as "rolling in $d_\mu^{\pi_\omega^\star}$ with probability $\beta$". We refer to this approach, which consists of *rolling in a similar optimal state visitation distribution of another context*, as ROLLIN.

We illustrate the intuition of ROLLIN in Figure 1. More specifically, we adopt the contextual MDP formulation, where we assume each MDP $\mathcal{M}_\omega$ is uniquely defined by a context $\omega$ in the context space $\mathcal{W} \subset \mathbb{R}^n$, and we are given a curriculum $\{\omega_k\}_{k=0}^K$, with the last MDP $\mathcal{M}_{\omega_K}$ being the MDP of interest. To show our main results, we only require a Lipschitz continuity assumption on $r_\omega$ w.r.t. $\omega$ and some mild regularity conditions on the curriculum $\{\omega_k\}_{k=0}^K$. We show that learning $\pi_K^\star$ by recursively rolling in with a near-optimal policy for $\omega_k$ to construct the initial distribution $\mu_{k+1}$ for the next context $\omega_{k+1}$, is provably more efficient than learning $\pi_{\omega_K}^\star$ from scratch. In particular, we show that when an appropriate sequence of contexts is selected, we can reduce the iteration and sample complexity bounds of entropy-regularized softmax policy gradient (with an inexact stochastic estimation of the gradient) from an original exponential dependency on the state space size, as suggested by Ding et al. (2021), to a polynomial dependency. We also prescribe a practical implementation of ROLLIN.

In summary, our contributions can be stated as follows. We first provide a theoretical method (ROLLIN) that facilitates single-task policy learning by recasting it as a multi-task

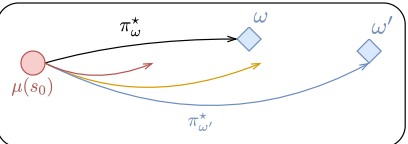

Learning $\pi_{\omega'}^\star$ from scratch

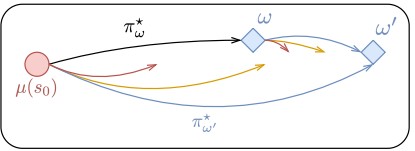

Learning $\pi_{\omega'}^\star$ with ROLLIN

Figure 1 Illustration of ROLLIN. The red circle represents the initial state distribution. **The dark curve** represents the optimal policy w.r.t. $\omega$. **The blue diamonds** represent the optimal state distributions $d_\mu^{\pi_\omega^\star}, d_\mu^{\pi_{\omega'}^\star}$ of $\omega, \omega'$ respectively. For learning $\pi_{\omega'}^\star$, when the optimal policy $\pi_\omega^\star$ of a similar context is known, ROLLIN rolls in the optimal policy of a near by context $\omega$ with probability $\beta$.

problem under entropy-regularized softmax policy gradient (PG), which reduces the exponential complexity bound of the entropy-regularized PG to a polynomial dependency on $S$. Last but not least, we also provide a deep RL implementation of ROLLIN and demonstrate adding ROLLIN improves performance in simulated goal-reaching tasks with an oracle curriculum and a non-oracle curriculum learned from MEGA (Pitis et al., 2020), as well as several standard Mujoco locomotion tasks inspired by meta RL (Clavera et al., 2018).

## 2 RELATED WORK

**Convergence of policy gradient methods.** Theoretical analysis of policy gradient methods has a long history (Williams, 1992; Sutton et al., 1999; Konda and Tsitsiklis, 1999; Kakade and Langford, 2002; Peters and Schaal, 2008). Motivated by the recent empirical success (Schulman et al., 2015; 2017) in policy gradient (PG) methods, the theory community has extensively studied the convergence of PG in various settings (Fazel et al., 2018; Agarwal et al., 2021; 2020; Bhandari and Russo, 2019; Mei et al., 2020; Zhang et al., 2020b; Agarwal et al., 2020; Zhang et al., 2020a; Li et al., 2021; Cen et al., 2021; Ding et al., 2021; Yuan et al., 2022; Moskovitz et al., 2022). Agarwal et al. (2021) established the asymptotic global convergence of policy gradient under different policy parameterizations. We extend the result of entropy regularized PG with stochastic gradient (Ding et al., 2021) to the contextual MDP setting. In particular, our contextual MDP setting reduces the exponential state space dependency w.r.t. the iteration number and per iteration sample complexity suggested by Ding et al. (2021) to a polynomial dependency. We shall also clarify that there is much existing convergence analysis on other variants of PG that produce an iteration number that does not suffer from an exponential state space dependency (Agarwal et al., 2021; Mei et al., 2020), but they assume access to the *exact* gradient during each update of PG, while we assume a stochastic estimation of the gradient, which is arguably more practical.

**Contextual MDPs.** Contextual MDPs (or MDPs with side information) have been studied extensively in the theoretical RL literature (Abbasi-Yadkori and Neu, 2014; Hallak et al., 2015; Dann et al.,

2019; Jiang et al., 2017; Modi et al., 2018; Sun et al., 2019; Dann et al., 2019; Modi et al., 2020). We analyze the iteration complexity and sample complexity of (stochastic) policy gradient methods, which is distinct from these prior works that mainly focus on regret bounds (Abbasi-Yadkori and Neu, 2014; Hallak et al., 2015; Dann et al., 2019) and PAC bounds (Jiang et al., 2017; Modi et al., 2018; Sun et al., 2019; Dann et al., 2019; Modi et al., 2020). Several works assumed linear transition kernel and reward model (or generalized linear model (Abbasi-Yadkori and Neu, 2014)) with respect to the context (Abbasi-Yadkori and Neu, 2014; Modi et al., 2018; Dann et al., 2019; Modi et al., 2020; Belogolovsky et al., 2021). These assumptions share similarity to our assumptions — we have a weaker Lipschitz continuity assumption with respect to the context space (since linear implies Lipschitz) on the reward function and a stronger shared transition kernel assumption.

**Exploration.** A number of prior works have shown that one can reduce the complexity of learning an optimal policy with effective exploration methods (Azar et al., 2017; Jin et al., 2018; Du et al., 2019; Misra et al., 2020; Agarwal et al., 2020; Zhang et al., 2020d). The computational efficiency suggested by our work is different from some of the aforementioned prior methods that rely on adding exploration bonus for policy cover (Azar et al., 2017; Jin et al., 2018; Agarwal et al., 2020; Zhang et al., 2020d), as we assume access to a "good" curriculum which ensures the optimal policy defined by the next context is not too different from the optimal policy of the current context. The key intuition of our work is more related to the strategic exploration in the block MDP setting (Du et al., 2019; Misra et al., 2020). The block MDP setting assumes the state space can be encoded into a feature space with a sequential block structure and explores the next block in the feature space after having a policy that covers the previous blocks. Such sequential exploration procedure is similar to ROLLIN, as our method also considers solving the multi-task contextual MDPs in a sequential perspective, but ROLLIN focuses on learning the optimal policy of the next context once we have a near-optimal policy of the current context.

**Curriculum learning in reinforcement learning.** Curriculum learning is a powerful idea that has been widely used in RL (Florensa et al., 2017; Kim and Choi, 2018; Omidshafiei et al., 2019; Ivanovic et al., 2019; Akkaya et al., 2019; Portelas et al., 2020; Bassich et al., 2020; Fang et al., 2020; Klink et al., 2020; Dennis et al., 2020; Parker-Holder et al., 2022; Liu et al., 2022) (also see (Narvekar et al., 2020) for a detailed survey). Although curricula formed by well-designed reward functions (Vinyals et al., 2019; OpenAI, 2018; Berner et al., 2019; Ye et al., 2020; Zhai et al., 2022) are usually sufficient given enough domain knowledge, tackling problems with limited domain knowledge requires a more general approach where a suitable curriculum is automatically formed from a task space. In the goal-conditioned reinforcement learning literature, this corresponds to automatic goal proposal mechanisms (Florensa et al., 2018; Warde-Farley et al., 2018; Sukhbaatar et al., 2018; Ren et al., 2019; Ecoffet et al., 2019; Hartikainen et al., 2019; Pitis et al., 2020; Zhang et al., 2020c; OpenAI et al., 2021; Zhang et al., 2021). The practical instantiation of this work is also similar to Bassich et al. (2020); Liu et al. (2022), where a curriculum is adopted for learning a progression of a set of tasks.

**Learning conditional policies in multi-task RL.** Multi-task RL (Tanaka and Yamamura, 2003) approaches usually learn a task-conditioned policy that is shared across different tasks (Rusu et al., 2015; Rajeswaran et al., 2016; Andreas et al., 2017; Finn et al., 2017; D'Eramo et al., 2020; Yu et al., 2020a; Ghosh et al., 2021; Kalashnikov et al., 2021). Compared to learning each task independently, joint training enjoys the sample efficiency benefits from sharing the learned experience across different tasks as long as the policies generalize well across tasks. To encourage generalization, it is often desirable to condition policies on low dimensional feature representations that are shared across different tasks instead (e.g., using variational auto-encoders (Nair et al., 2018; Pong et al., 2019; Nair et al., 2020) or variational information bottleneck (Goyal et al., 2019; 2020; Mendonca et al., 2021)). The idea of learning contextual policies has also been discussed in classical adaptive control literature (Sastry et al., 1990; Tao, 2003; Landau et al., 2011; Åström and Wittenmark, 2013; Goodwin and Sin, 2014). Different from these prior works which have been mostly focusing on learning policies that can generalize across different tasks, our work focuses on how the near-optimal policy from a learned task could be used to help the learning of a similar task.

## 3 PRELIMINARIES

We consider the contextual MDP setting, where a contextual MDP $\mathcal{M}_{\mathcal{W}} = (\mathcal{W}, \mathcal{S}, \mathcal{A}, \boldsymbol{P}, r_\omega, \gamma, \rho)$ consists of a context space $\mathcal{W}$, a state space $\mathcal{S}$, an action space $\mathcal{A}$, a transition dynamic function $\boldsymbol{P} : \mathcal{S} \times \mathcal{A} \to \mathcal{P}(\mathcal{S})$ (where $\mathcal{P}(X)$ denotes the set of all probability distributions over set $X$), a context-conditioned reward function $r : \mathcal{W} \times \mathcal{S} \times \mathcal{A} \to [0, 1]$, a discount factor $\gamma \in (0, 1]$, and an initial state distribution of interest $\rho$. For convenience, we use $S = |\mathcal{S}|, A = |\mathcal{A}|$ to

denote the number of states and actions. While some contextual MDP formulations (Hallak et al., 2015) have context-conditioned transition dynamics and reward functions, we consider the setting where only the reward function can change across contexts. We denote $r_\omega$ as the reward function conditioned on a fixed $\omega \in \mathcal{W}$ and $\mathcal{M}_\omega = (\mathcal{S}, \mathcal{A}, \boldsymbol{P}, r_\omega, \gamma, \rho)$ as the MDP induced by such fixed reward function. We use $\pi(a|s) : \mathcal{S} \to \mathcal{P}(\mathcal{A})$ to denote a policy and we adopt the softmax parameterization: $\pi_\theta(a|s) = \frac{\exp[\theta(s,a)]}{\sum_{a'} \exp[\theta(s,a')]}$, where $\theta : \mathcal{S} \times \mathcal{A} \mapsto \mathbb{R}$. We use $d_\rho^\pi(s) := (1 - \gamma) \sum_{t=0}^\infty \gamma^t \mathbb{P}^\pi(s_t = s | s_0 \sim \rho)$ to denote the discounted state visitation distribution and $V_\omega^\pi := \mathbb{E}\left[\sum_{t=0}^\infty \gamma^t r_\omega(s_t, a_t)\right] + \alpha \mathbb{H}(\rho, \pi)$ to denote the entropy regularized discounted return on $\mathcal{M}_\omega$, where $\mathbb{H}(\rho, \pi) := \mathbb{E}_{s_0 \sim \rho, a_h \sim \pi(\cdot|s_h)}\left[\sum_{h=0}^\infty -\gamma^h \log \pi(a_h|s_h)\right]$ is the discounted entropy term. We use $\pi_\omega^\star := \arg\max_\pi V_\omega^\pi$ to denote an optimal policy that maximizes the discounted return under $\mathcal{M}_\omega$. We assume all the contextual reward functions are bounded within $[0, 1]$: $r_\omega(s, a) \in [0, 1]$, $\forall \omega \in \Omega, \forall(s, a) \in \mathcal{S} \times \mathcal{A}$. Similarly to previous analysis Agarwal et al. (2021); Mei et al. (2020); Ding et al. (2021), we assume the initial distribution $\rho$ for PG or stochastic PG satisfies $\rho(s) > 0, \forall s \in \mathcal{S}$.

### 3.1 Assumptions

Given a curriculum $\{\omega_k\}_{k=0}^K$, where the last context $\omega_K$ defines $\mathcal{M}_{\omega_K}$, the MDP of interest, our goal is to show that reformulating $\mathcal{M}_{\omega_K}$ into a multi-task problem $\{\mathcal{M}_{\omega_k}\}_{k=0}^K$, enjoys better computational complexity and sample complexity than solving the single-task problem $\mathcal{M}_{\omega_K}$ from scratch. As we will show in Section 4, if we have a "good" curriculum $\{\omega_k\}_{k=0}^K$ such that the optimal policies $\pi_{\omega_k}^\star, \pi_{\omega_{k+1}}^\star$ w.r.t. two consecutive contexts $\omega_k, \omega_{k+1}$ are "close enough" to each other, then using an $\varepsilon$-optimal policy of $\omega_k$ as an initialization allows us to directly start from the near optimal regime of $\omega_{k+1}$, hence only requiring polynomial complexity to learn $\pi_{\omega_{k+1}}^\star$. Formally, we use the two following assumptions to characterize the good properties of our curriculum.

**Assumption 3.1 (Lipschitz reward in the context space)** *The reward function is Lipschitz continuous w.r.t. the context:* $\max_{s,a} |r_\omega(s, a) - r_{\omega'}(s, a)| \leq L_r \|\omega - \omega'\|_2, \forall \omega, \omega' \in \mathcal{W}$.

**Assumption 3.2 (Similarity between two Consecutive Contexts)** *We assume the given curriculum* $\{\omega_k\}_{k=0}^K$ *satisfies* $\max_{0 \leq k \leq K-1} \|\omega_{k+1} - \omega_k\|_2 \leq O\left(S^{-2}\right)$, *and we have access to a near-optimal initialization* $\theta_0^{(0)}$ *for learning* $\pi_{\omega_0}^\star$ *(formally define in Section 4.2).*

Intuitively, Assumption 3.1 defines the similarity between two tasks via a Lipschitz continuity in the context space, similar Lipschitz assumption also appears in Abbasi-Yadkori and Neu (2014); Modi et al. (2018); Dann et al. (2019); Modi et al. (2020); Belogolovsky et al. (2021). Assumption 3.1 and 3.2 together quantify the maximum difference between two consecutive tasks $\mathcal{M}_{\omega_{k-1}}, \mathcal{M}_{\omega_k}$, in terms of the maximum difference between their reward function. Assumption 3.1 and 3.2 also play a crucial role in reducing the exponential complexity to a polynomial one, and we will briefly discuss the intuition in the next section.

### 3.2 Prior Analysis on PG with stochastic gradient

Ding et al. (2021) proposed a two-phased PG convergence analysis framework with a stochastic gradient. In particular, the author demonstrates that with high probability, stochastic PG with arbitrary initialization achieves an $\varepsilon$-optimal policy can be achieved with an iteration number of $T_1, T_2$ and per iteration sample complexity of $B_1, B_2$ in two separate phases where $T_1 = \widetilde{\Omega}(S^{2S^3}), T_2 = \widetilde{\Omega}(S^{3/2})$ ($\widetilde{\Omega}(\cdot)$ suppresses the $\log S$ and terms that do not contain $S$) and $B_1 = \widetilde{\Omega}(S^{2S^3}), B_2 = \widetilde{\Omega}(S^5)$, respectively, and PG enters phase 2 only when the updating policy becomes $\varepsilon_0$-optimal, where $\varepsilon_0$ is a term depending on $S$ (formally defined by (19) in Appendix A.3). For completeness, we restate the main theorem of Ding et al. (2021) in Theorem A.2, provide the details of such dependencies on $S$ are provided in Corollary A.3, and describes the two phase procedure in Algorithm 4. The main implication of these two-phase results is that, if we want to apply PG to learn an optimal policy from an arbitrary initialization, we suffer from an exponential number and sample complexity, unless the initialization is $\varepsilon_0$-optimal. In the next section, we will discuss how Assumption 3.1 and 3.2 enables an $\varepsilon_0$-optimal initialization for every $\omega_k$ hence reducing the exponential complexity to a polynomial one.

## 4 Theoretical Analysis

In this section, we introduce ROLLIN, a simple algorithm that accelerates policy learning under the contextual MDP setup by bootstrapping new context learning with a better initial distribution (Algorithm 1). We also provide the total complexity analysis of applying ROLLIN to stochastic PG for achieving an $\varepsilon$-optimal policy.

### 4.1 ROLLIN

The theoretical version of ROLLIN is provided in Algorithm 1. Our intuition behind ROLLIN is that when two consecutive contexts in the curriculum $\{\omega_k\}_{k=1}^K$ are close, their optimal parameters $\theta_{\omega_{k-1}}^\star, \theta_{\omega_k}^\star$ should be close to each other. Let $\theta_t^{(k)}$ denote the parameters at the $t^{\text{th}}$ iteration of stochastic PG for learning $\theta_{\omega_k}^\star$, if we initialize the parameter $\theta_1^{(k)}$ as the optimal parameter of the previous context $\theta_{\omega_{k-1}}^\star$ (line 5 in Algorithm 1), and set the initial distribution $\mu_k$ as a mixture between the optimal state visitation distribution of the previous context $d_{\mu_{k-1}}^{\pi_{\omega_{k-1}}^\star}$ and the original distribution of interest $\rho$ with ratio $\beta$ (line 6 in Algorithm 1), such that

$$\mu_k = \beta d_{\mu_{k-1}}^{\pi_{\omega_{k-1}}^\star} + (1-\beta)\rho, \tag{1}$$

then stochastic PG enjoys a faster convergence rate. This happens because setting $\theta_1^{(k)} = \theta_{k-1}^\star$ ensures a near-optimal initialization for $\theta_1^{(k)}$, and setting $\mu_k$ by mixing $d_{\mu_{k-1}}^{\pi_{\omega_{k-1}}^\star}$ and $\rho$ further improves the rate of convergence by decreasing the density mismatch ratio $\left\| d_{\mu_k}^{\pi_{\omega_k}^\star}/\mu_k \right\|_\infty$ (a term with an $S$ dependency that influences the rate of convergence).

---

**Algorithm 1** Provably Efficient Learning via ROLLIN

1: **Input:** $\rho$, $\{\omega_k\}_{k=0}^K$, $\mathcal{M}_{\mathcal{W}}$, $\beta \in (0,1)$, $\theta_0^{(0)}$.
2: Initialize $\mu_0 = \rho$.
3: Run stochastic PG (Algorithm 4) with initialization $\theta_0^{(0)}$, $\mu_0$, $\mathcal{M}_{\omega_0}$ and obtain $\theta_{\omega_0}^\star$.
4: **for** $k = 1, \ldots, K$ **do**
5:      Set $\theta_1^{(k)} = \theta_{\omega_{k-1}}^\star$.      $\triangleright \pi_{\omega_{k-1}}^\star = \pi_{\theta_{\omega_{k-1}}^\star}$ is the optimal policy of $\omega_{k-1}$.
6:      Set $\mu_k = \beta d_{\mu_{k-1}}^{\pi_{\omega_{k-1}}^\star} + (1-\beta)\rho$.      $\triangleright \mu_k$ is the initial distribution for learning $\theta_{\omega_k}^\star$.
7:      Run stochastic PG (Algorithm 4) with initialization $\theta_1^{(k)}$, $\mu_k$, $\mathcal{M}_{\omega_k}$ and obtain $\theta_{\omega_k}^\star$.
8: **end for**
9: **Output:** $\theta_{\omega_K}^\star$

---

### 4.2 MAIN RESULTS

We now discuss how to use a sequence of contexts to learn the target context $\omega_K$ with provable efficiency given the optimal policy $\pi_{\omega_0}^\star$ of the initial context $\omega_0$, without incurring an exponential dependency on $S$ (as mentioned in Section 3.2). Our polynomial complexity comes as a result of enforcing an $\varepsilon_0$-optimal initialization ($\varepsilon_0$ is the same as Section 3.2 and (19)) for running stochastic PG (line 6 of Algorithm 1). Hence, stochastic PG directly enters phase 2, with a polynomial dependency in $S$.

Our main results consist of two parts. We first show that when two consecutive contexts $\omega_{k-1}, \omega_k$ are close enough to each other, using ROLLIN for learning $\theta_k^\star$ with initialization $\theta_1^{(k)} = \theta_{\omega_{k-1}}^\star$ and applying an initial distribution $\mu_k = \beta d_{\mu_{k-1}}^{\pi_{\omega_{k-1}}^\star} + (1-\beta)\rho$ improves the convergence rate. Specifically, the iteration number and complexity for learning $\theta_{\omega_k}^\star$ from $\theta_{\omega_{k-1}}^\star$ is stated as follows:

**Theorem 4.1 (Complexity of Learning the Next Context)** *Consider the context-based stochastic softmax policy gradient (line 7 of Algorithm 1), suppose Assumption 3.1 and Assumption 3.2 hold, then the iteration number of obtaining an $\varepsilon$-optimal policy for $\omega_k$ from $\theta_{\omega_{k-1}}^\star$ is $\widetilde{\Omega}\left(S^{3/2}\right)$ and the per iteration sample complexity is $\widetilde{\Omega}\left(\frac{L_r}{\alpha(1-\beta)}S^3\right)$.*

In other words, Theorem 4.1 shows that when $\omega_{k-1}, \omega_k$ are close enough, ROLLIN reduces the iteration and sample complexity from an exponential dependency of $\widetilde{\Omega}(S^{2S^3})$ to an iteration number of $\widetilde{\Omega}(S^{3/2})$ and per iteration sample complexity of $\widetilde{\Omega}(S^3)$. It is worth noting that the theorem above only address the iteration numbers and sample complexity for learning $\theta_{\omega_k}^\star$ from $\theta_{\omega_{k-1}}^\star$.

Theorem 4.3 provides the total complexity for learning $\theta_{\omega_K}^\star$ from $\theta_0^{(0)}$ via recursively apply the results in Theorem 4.1. Before introducing Theorem 4.3, we first provide a criteria for the desired initialization of $\theta_0^{(0)}$.

**Definition 4.2 (Near-optimal Initialization)** *We say $\theta_0$ is a near-optimal initialization for learning $\theta_\omega^\star$ if $\theta_0$ satisfies $V_\omega^{\pi_\omega^\star}(\rho) - V_\omega^{\pi_{\theta_0}}(\rho) < \varepsilon_0$.*

Note that in the above definition, $\pi^\star_{\omega_k}$ represents the optimal policy of $\omega_k$, and $V^\pi_{\omega_k}$ represents value function of context $\omega_k$ under policy $\pi$. Now we introduce the results for the overall complexity:

**Theorem 4.3 (Main Results: Total Complexity of ROLLIN)** *Suppose Assumption 3.1 and 3.2 hold, and $\theta_0^{(0)}$ is an near-optimal initialization, then the total number of iteration of learning $\pi^\star_{\omega_K}$ using Algorithm 1 is $\Omega(KS^{3/2})$ and the per iteration is $\widetilde{\Omega}\left(S^3\right)$, with high probability.*

A direct implication of Theorem 4.3 is that, with a curriculum $\{\omega_k\}_{k=0}^K$ satisfying Assumption 3.1 and 3.2, one can reduce the daunting exponential dependency on $S$ caused by poor initialization to a polynomial dependency on $S$. Admittedly the state space $S$ itself is still large in practice, but reducing the state space $S$ itself requires extra assumptions on $\mathcal{S}$ which is beyond the scope of this work. We will provide a sketch proof of Theorem 4.1 and 4.3 in the next subsection and leave all the details proof to Appendix A.4, A.5 respectively.

### 4.3 PROOF SKETCH

**Sketch proof of Theorem 4.1** The key insight for proving Theorem 4.1 is to show that in MDP $\mathcal{M}_{\omega_k}$, the value function w.r.t. $\pi^\star_{\omega_k}, \pi^\star_{\omega_{k-1}}$ can be bounded by the $\ell^2$ norm between $\omega_k$ and $\omega_{k-1}$. In particular, we prove such a relation in Lemma A.5:

$$V_{\omega_k}^{\pi^\star_{\omega_k}}(\rho) - V_{\omega_k}^{\pi^\star_{\omega_{k-1}}}(\rho) \leq \frac{2L_r \|\omega_k - \omega_{k-1}\|_2}{(1-\gamma)^2}. \tag{2}$$

By setting $\theta_1^{(k)} = \theta^\star_{\omega_{k-1}}$, Equation (2) directly implies $V_{\omega_k}^{\pi^\star_{\omega_k}}(\rho) - V^{\theta_1^{(k)}}(\rho) \leq \frac{2L_r \|\omega_k - \omega_{k-1}\|_2}{(1-\gamma)^2}$. As suggested by Ding et al. (2021) stochastic PG can directly start from stage 2 with polynomial complexity of $T_2 = \widetilde{\Omega}(S^{3/2})$, $B_2 = \widetilde{\Omega}(S^5)$, if $V_{\omega_k}^{\pi^\star_{\omega_k}}(\rho) - V^{\theta_1^{(k)}}(\rho) \leq \varepsilon_0$, where $\varepsilon_0$ (formally defined in Equation (19) in Appendix A.3) is a constant satisfying $\varepsilon_0 = O(S^{-2})$. Hence, by enforcing two consecutive contexts to be close enough $\|\omega_k - \omega_{k-1}\|_2 \leq O(S^{-2})$, we can directly start from a near-optimal initialization with polynomial complexity w.r.t. $S$. This largely explains the intuition behind the initialization in line 5 of ROLLIN: $\theta_1^{(k)} = \theta^\star_{\omega_{k-1}}$. It is worth highlighting that the per iteration sample complexity $B_2$ shown by Ding et al. (2021) scales as $\widetilde{\Omega}(S^5)$, while our result in Theorem 4.1 only requires a smaller sample complexity of $\widetilde{\Omega}(S^3)$. Such an improvement in the sample complexity comes from line 6 of ROLLIN: $\mu_k = \beta d_{\mu_{k-1}}^{\pi^\star_{\omega_{k-1}}} + (1-\beta)\rho$. Intuitively, setting $\mu_k$ as $\beta d_{\mu_{k-1}}^{\pi^\star_{\omega_{k-1}}} + (1-\beta)\rho$ allows us to provide an upper bound on the density mismatch ratio:

$$\left\| d_{\mu_k}^{\pi^\star_{\mu_k}} / \mu_k \right\|_\infty \leq \widetilde{\Omega}\left(\frac{L_r}{\alpha(1-\beta)}\Delta^k_\omega S\right), \tag{3}$$

where $\Delta^k_\omega = \max_{1 \leq i \leq k} \|\omega_i - \omega_{i-1}\|_2$. Since the sample complexity $B_2$ (provided in Corollary A.3) contains one multiplier of $\left\| d_{\mu_k}^{\pi^\star_{\mu_k}} / \mu_k \right\|_\infty$, setting $\Delta^k_\omega = O(S^{-2})$ immediately reduces the complexity by an order of $S^2$. The proof of the upper bound of the density mismatch ratio (Equation 3) is provided in Lemma A.1.

**Sketch proof of Theorem 4.3** We obtain Theorem 4.3 by recursively applying Theorem 4.1. More precisely, we use induction to show that, if we initialize the parameters of the policy as $\theta_1^{(k)} = \theta^\star_{\omega_{k-1}}$, when $t = \widetilde{\Omega}(S^{3/2})$, $\forall k \in [K]$, we have $V_{\omega_k}^{\pi^\star_{\omega_k}}(\rho) - V_{\omega_k}^{\pi_{\theta_t^{(k-1)}}}(\rho) < \varepsilon_0$. Hence, for any context $\omega_k, k \in [K]$, initializing $\theta_1^{(k)} = \theta_t^{(k-1)}$ from learning $\pi^\star_{\omega_{k-1}}$ via stochastic PG after $t = \Omega(S^{3/2})$ iteration, $\theta_1^{(k)}$ will directly start from the efficient phase 2 with polynomial complexity. Hence, the total iteration number for learning the $\theta^\star_K$ is $\Omega(KS^{3/2})$, and the per iteration sample complexity $\widetilde{\Omega}\left(S^3\right)$ remains the same as Theorem 4.1.

## 5 PRACTICAL IMPLEMENTATION OF ROLLIN

In this section, we describe the practical implementation of ROLLIN using Soft-Actor-Critic (Haarnoja et al., 2018) algorithm when an oracle curriculum is available. SAC can be seen as a variant of entropy-regularized stochastic PG with the addition of the critics to reduce the gradient variance. Recall that in the theoretical analysis, we learn a separate policy for each context that can start from the near-optimal state distribution of the previous context to achieve a good return under the current

context. However, in practice, we usually would want to have a policy that can directly start from the initial distribution $\rho$ to obtain a good return for the final context $\omega_K$. In order to learn such a policy, we propose to have two context-conditioned RL agents training in parallel, where the first agent $\pi_{\text{main}}$ is the main agent that eventually will learn to achieve a good return from $\rho$, and the second agent $\pi_{\text{exp}}$ is an exploration agent that learns to achieve a good return under the current context from the near-optimal state density of the previous context. Another purpose of the exploration agent (as the name suggests) is to provide a better exploration experience for the main agent to learn the current context better. This is made convenient by using an off-policy RL agent where the main agent can easily learn from data that is not on-policy.

Specifically, for each episode, there is a probability of $\beta$ where we run the main agent conditioned on the previous context for a random number of steps until we switch to the exploration agent to collect experience for the current context until the episode ends. Otherwise, we directly run the main agent for the entire episode. Both agents are trained to maximize the return under the current context. Whenever the average return of the last 10 episodes exceed a performance threshold $R$, we immediately switch to the next context and re-initialize the exploration agent and its replay buffer. A high-level description is available in Algorithm 2 (a more detailed version in Algorithm 8).

---

**Algorithm 2** Practical Implementation of ROLLIN

1: **Input:** $\{\omega_k\}_{k=0}^K$: input curriculum, $R$: near-optimal threshold, $\beta$: roll-in ratio, $H$: horizon, $\gamma$: discount factor.
2: Initialize $\mathcal{D} \leftarrow \emptyset, \mathcal{D}_{\text{exp}} \leftarrow \emptyset, k \leftarrow 0$, and two SAC agents $\pi_{\text{main}}$ and $\pi_{\text{exp}}$.
3: **for** each episode **do**
4:     **if** average return of the last 10 episodes under context $\omega_k$ is greater than $R$ **then**
5:         $k \leftarrow k + 1, \mathcal{D}_{\text{exp}} \leftarrow \emptyset$, and re-initialize the exploration agent $\pi_{\text{exp}}$
6:     **end if**
7:     **if** $k > 0$ and with probability of $\beta$ **then**
8:         $h \sim \text{Geom}(1 - \gamma)$ (truncated at $H$)
9:         run $\pi_{\text{main}}(a|s, \omega_{k-1})$ from the initial state for $h$ steps and switch to $\pi_{\text{exp}}(a|s, \omega_k)$ until the episode ends to obtain trajectory $\tau_{0:H} = \{s_0, a_0, r_0, s_1, a_1, \cdots, s_H\}$.
10:         record $\tau_{0:H}$ in $\mathcal{D}$, and $\tau_{h:H}$ in $\mathcal{D}_{\text{exp}}$.
11:     **else**
12:         run $\pi_{\text{main}}(a|s, \omega_k)$ to obtain trajectory $\tau_{0:H}$ and record $\tau_{0:H}$ in $\mathcal{D}$.
13:     **end if**
14:     at each environment step during the episode, update $\pi_{\text{main}}$ using $\mathcal{D}$ and $\pi_{\text{exp}}$ using $\mathcal{D}_{\text{exp}}$.
15: **end for**
16: **Output:** $\pi_{\text{main}}$

---

## 6 EXPERIMENTAL RESULTS

While the focus of our work is on developing a provably efficient approach to curriculum learning, we also conduct an experimental evaluation of our practical implementation of ROLLIN with soft actor-critic (SAC) (Haarnoja et al., 2018) as the RL algorithm on several continuous control tasks including goal reaching with an oracle curriculum and a learned curriculum, as well as non-goal tasks.

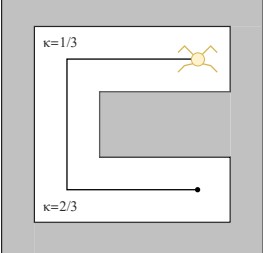

### 6.1 GOAL REACHING WITH AN ORACLE CURRICULUM

We adopt the `antmaze-umaze` environment (Fu et al., 2020) for evaluating the performance of ROLLIN in goal-reaching tasks. In the oracle curriculum case, we use a hand-crafted path of contexts, where each context specifies the location that the ant needs to reach (as shown in Figure 2).[1] We consider a path of contexts $\omega(\kappa)$ parameterized by $\kappa \in [0, 1]$ where $\omega(0) = \omega_0$ and $\omega(1) = \omega_K$, and propose contexts along the path with a fixed step size $\Delta$. See Appendix E.1 for more implementation details.

Figure 2 Oracle curriculum of desired goals on `antmaze-umaze`.

We combine ROLLIN with a variety of prior methods, and we evaluate the following conditions: (1) standard goal reaching; (2) goal reaching with goal relabeling (Andrychowicz et al., 2017);

---

[1]Note that in general, ROLLIN is *not* specific to goal-conditioned RL and could in principle utilize any type of context-parameterized reward function that satisfies its assumptions.

(3) goal reaching with Go-Explore (Ecoffet et al., 2019). For goal relabeling, we adopt a similar relabeling technique as Pitis et al. (2020), where each mini batch contains $1/3$ original transitions, $1/3$ transitions with future state relabeling, and $1/3$ transitions with next state relabeling. We implemented the Go-Explore method by adding an additional standard Gaussian exploration noise (multiplied by a constant factor) to the agent for learning the next goal $\omega(k+1)$, once it reaches the current goal $\omega(k)$. We empirically observed that sampling the replay buffer from a geometric distribution with $p = 10^{-5}$ (more recent transitions are sampled more frequently) improves the overall performance. Hence, in all future experiments, we compare the performance of ROLLIN with classic uniform sampling and the new geometric sampling. We demonstrate how adjusting the ROLLIN parameter ($\beta = 0.1, 0.2, 0.5, 0.75, 0.9$) impacts the learning speed on three different step sizes $\Delta = \frac{1}{24}, \frac{1}{18}, \frac{1}{12}$. We expect coarser curricula with larger steps to be more difficult.

**Main comparisons.** We first provide an overview experiments that compares ROLLIN with a fixed $\beta = 0.1$ on different step sizes $\Delta$ in different settings. In each case, we compare the prior method (vanilla, relabeled, or Go-Explore) with and without the addition of ROLLIN. As shown in Table 1, ROLLIN improves the largest value of $\kappa$ reached by the agent in most presented settings (except Go-Explore with $\Delta = 1/12$). This result suggests that ROLLIN facilitates goal-conditioned RL with a curriculum, as we only update the learning progress $\kappa$ to $\kappa + \Delta$ when the return of the current policy reaches a certain threshold $R$ (See detailed update of $\kappa$ in Algorithm 2). Note that $\beta = 0.1$ does not always produce the best result, we will provide more results comparing different $\beta$s in different settings later in this section, and we leave all the learning curves and detailed tables to Appendix F.1. Note that we do not include the results of directly learning the last context in the `antmaze-umaze` environment because the agent cannot reach the goal without the aid of a curriculum, which is corroborated by Pitis et al. (2020).

| Setting | Method | w/o Geometric Sampling | | w/ Geometric Sampling | |
|---|---|---|---|---|---|
| | | $\Delta = 1/24$ | $\Delta = 1/12$ | $\Delta = 1/24$ | $\Delta = 1/12$ |
| Vanilla | Baseline | $0.40 \pm 0.02$ | $0.36 \pm 0.00$ | $0.82 \pm 0.08$ | $0.38 \pm 0.03$ |
| | ROLLIN | $0.49 \pm 0.04$ | $0.44 \pm 0.01$ | $0.92 \pm 0.02$ | $0.55 \pm 0.04$ |
| Relabeling | Baseline | $0.89 \pm 0.03$ | $0.66 \pm 0.04$ | $0.76 \pm 0.02$ | $0.72 \pm 0.03$ |
| | ROLLIN | $0.91 \pm 0.03$ | $0.74 \pm 0.01$ | $0.78 \pm 0.01$ | $0.73 \pm 0.00$ |
| Go-Explore | Baseline | $0.37 \pm 0.02$ | $0.38 \pm 0.01$ | $0.82 \pm 0.07$ | $0.42 \pm 0.03$ |
| Noise = 0.1 | ROLLIN | $0.52 \pm 0.07$ | $0.38 \pm 0.01$ | $0.95 \pm 0.02$ | $0.43 \pm 0.02$ |

Table 1 Learning progress $\kappa$ at 3 million environment steps with varying curriculum step size $\Delta$ of different settings of goal reaching in `antmaze-umaze`. We pick $\beta = 0.1$ for all experiments using ROLLIN, the results of using other $\beta$s, $\Delta$s, and exploration noise can be found in Table 5, 6, 7 in Appendix F.1. The standard error is computed over 8 random seeds.

## 6.2 GOAL REACHING WITH A LEARNED CURRICULUM

In this section, we focus on the setting where an oracle curriculum is not provided. In particular, we show that ROLLIN can improve MEGA (Pitis et al., 2020), an existing automated goal curriculum generation method (Figure 3). MEGA proposes intrinsic goals that are achievable, but appear rarely in the replay buffer. Since there is no clear notion of progression in the learned curriculum setting, we use the progression in the number of environment steps where $\omega_k$ corresponds to the goals during $k \times 100K$ - $(k + 1) \times 100K$ steps. We follow the same high-level procedure as described in Algorithm 2 where we randomly roll in a policy that was trained on the previous context (in this case, it would be the policy checkpoint that was obtained after the last 100K-step learning segment). A detailed algorithm procedure is provided in Algorithm 9.

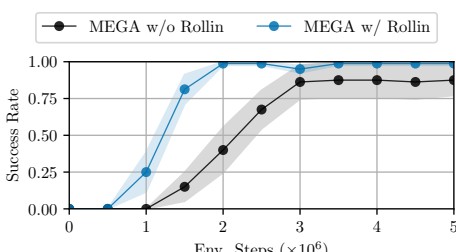

Figure 3 The success rate of goal reaching in `antmaze-umaze` with a curriculum generated by MEGA with $\beta = 0.5$. The success rate of learned policy on reaching the target goal is reported over 8 independent random seeds. 10 random trials for each seed are used to estimate the success rate. See Appendix E.2 for more details and results for other $\beta$s.

## 6.3 NON-GOAL REACHING TASKS

For the non-goal tasks, we choose a fixed contextual space with ten discrete contexts: $\kappa \in \{0.1, 0.2, \ldots, 1\}$, where each $\kappa$ uniquely determines the desired $x$-velocity of a locomotion agent in

the following environments: `walker2d`, `hopper`, `humanoid`, and `ant` in OpenAI gym (Brock-man et al., 2016). Our goal is to train an agent to move fast and stably. For each context $\omega(\kappa)$, we set the desired speed range to be $[\lambda\kappa, \lambda(\kappa + 0.1))$, where $\lambda$ is a parameter depending on the physics of the agent in different environments. When the $x$-velocity of an agent is within the desired velocity range, we set the `healthy_reward` to a higher value `healthy_reward_high`. Otherwise, we set the `healthy_reward` to a lower value `healthy_reward_low`. In each environment, we increase the task difficulty with later curriculum steps (larger $\kappa$), by increasing the near-optimal threshold $R(\kappa)$. Detailed parameters of the desired speed range $\lambda$, near optimal-threshold $R(\kappa)$, `healthy_reward_high`, and `healthy_reward_low` are provided in Appendix E.3.

**Main comparisons.** We first compare ROLLIN with a fixed $\beta = 0.1$ at different environment steps: $0.75 \times 10^6$, $1 \times 10^6$. In each case, we compare the learning progress $\kappa$, averaged $x$-velocity, and averaged return, with and without the addition of ROLLIN. Note that for the case without ROLLIN, we still provide the curriculum to the agent for training. As shown in Table 2, ROLLIN improves the largest learning progresses $\kappa$, average $x$-velocity, and average return in most presented settings. This result suggests that ROLLIN also can facilitate learning of non-goal tasks, as we only update the learning progress $\kappa$ to $\kappa + 0.1$ when the return of the current policy reaches a certain threshold $R(\kappa)$ (See detailed update of $\kappa$ in Algorithm 2). Note that $\beta = 0.1$ does not always produce the best result, we will provide more results comparing different $\beta$s in different settings later in this section, and we leave all the learning curves and detailed tables to Appendix F.3.

| Env. | Method | Step = $0.75 \times 10^6$ | | | Step = $1.0 \times 10^6$ | | |
| | | $\kappa$ | $x$-velocity | return | $\kappa$ | $x$-velocity | return |
|---|---|---|---|---|---|---|---|
| walker | Scratch | n/a | $2.90 \pm 0.59$ | $3937.6 \pm 120.6$ | n/a | $3.26 \pm 0.36$ | $4212.3 \pm 151.4$ |
| | Baseline | $0.89 \pm 0.03$ | $3.20 \pm 0.40$ | $3838.2 \pm 234.0$ | $0.92 \pm 0.03$ | $3.69 \pm 0.27$ | $4032.3 \pm 224.3$ |
| | ROLLIN | $0.90 \pm 0.02$ | $3.48 \pm 0.29$ | $3831.7 \pm 133.3$ | $0.94 \pm 0.03$ | $3.62 \pm 0.26$ | $4128.8 \pm 159.6$ |
| hopper | Scratch | n/a | $2.27 \pm 0.30$ | $3081.0 \pm 93.5$ | n/a | $2.52 \pm 0.12$ | $3073.2 \pm 137.7$ |
| | Baseline | $0.86 \pm 0.02$ | $2.48 \pm 0.14$ | $3325.8 \pm 100.6$ | $0.88 \pm 0.01$ | $2.58 \pm 0.16$ | $3386.2 \pm 124.7$ |
| | ROLLIN | $0.85 \pm 0.02$ | $2.54 \pm 0.17$ | $3339.7 \pm 151.4$ | $0.89 \pm 0.00$ | $2.65 \pm 0.15$ | $3421.9 \pm 109.8$ |
| humanoid | Scratch | n/a | $0.30 \pm 0.05$ | $2737.9 \pm 85.5$ | n/a | $0.37 \pm 0.05$ | $2763.8 \pm 96.5$ |
| | Baseline | $0.54 \pm 0.01$ | $0.32 \pm 0.05$ | $3007.5 \pm 176.6$ | $0.67 \pm 0.03$ | $0.39 \pm 0.05$ | $3017.2 \pm 169.0$ |
| | ROLLIN | $0.67 \pm 0.03$ | $0.39 \pm 0.06$ | $3151.0 \pm 171.6$ | $0.69 \pm 0.06$ | $0.46 \pm 0.09$ | $3173.6 \pm 238.3$ |
| ant | Scratch | n/a | $4.36 \pm 0.36$ | $3946.2 \pm 169.4$ | n/a | $4.45 \pm 0.38$ | $4277.9 \pm 120.0$ |
| | Baseline | $0.93 \pm 0.02$ | $4.30 \pm 0.37$ | $3151.0 \pm 224.4$ | $1.00 \pm 0.00$ | $4.29 \pm 0.51$ | $4248.5 \pm 88.6$ |
| | ROLLIN | $0.99 \pm 0.01$ | $4.50 \pm 0.30$ | $4242.1 \pm 89.5$ | $1.00 \pm 0.00$ | $4.66 \pm 0.30$ | $4473.0 \pm 102.2$ |

Table 2 Learning progress $\kappa$, average $x$-velocity, and average return at the 0.75 and 1.0 million environment steps in `walker`, `hopper`, `humanoid`, and `ant`. The average $x$-velocity and return are estimated using the last 50k time steps. "Scratch" shows the results of directly training the agent with the last context $\omega(1)$. "Baseline" indicates $\beta = 0$ where we provide the curriculum $\omega(\kappa)$ to the agent without using ROLLIN. We pick $\beta = 0.1$ for all experiments using ROLLIN, the results of using other $\beta$s can be found in Table 8, 9, 10 in Appendix F.3. The standard error is computed over 8 random seeds.

## 6.4 EXPERIMENTAL SUMMARY

In summary, we empirically demonstrated that ROLLIN improves the performance of goal-reaching tasks and non-goal tasks in different settings with a vast range of selection of $\beta$s. Although ROLLIN introduces an extra parameter $\beta$ for finetuning goal-reaching tasks with a curriculum, our extensive experiments suggest that one could expect improvement by choosing a constant $\beta = 0.1$ or 0.2. Still, for achieving the best performance in general-task, one shall still consider fine-tuning $\beta$.

## 7 DISCUSSION AND FUTURE WORK

We presented ROLLIN, a simple algorithm that accelerates curriculum learning under the contextual MDP setup by rolling in a near-optimal policy to bootstrap the learning of new nearby contexts with provable learning efficiency benefits. Theoretically, we show that ROLLIN attains polynomial sample complexity by utilizing adjacent contexts to initialize each policy. Since the key theoretical insight of ROLLIN suggests that one can reduce the density mismatch ratio by constructing a new initial distribution, it would be interesting to see how ROLLIN can affect other variants of convergence analysis of PG (e.g., NPG (Kakade, 2001; Cen et al., 2021) or PG in a feature space (Agarwal et al., 2021; 2020)). On the empirical side, our extensive experiments demonstrate that ROLLIN improves the empirical performance of various tasks beyond our theoretical assumptions, which reveals the potential of ROLLIN in other practical RL tasks with a curriculum. Our experiments also suggest that one could expect improvement by choosing a constant $\beta = 0.1$ or 0.2. Hence, another potential direction on the empirical side would be providing an automatic tuning method to select the best $\beta$.

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

# APPENDIX A  GENERALIZATION BETWEEN DIFFERENT TASKS IN THE CONTEXT SPACE

## A.1  SUMMARIES OF NOTATIONS AND ASSUMPTIONS

1. The maximum entropy RL (MaxEnt RL) objective with initial state distribution $\rho$ in reinforcement aims at maximizing (Equation 15 & 16 of Mei et al. (2020))

$$V^\pi(\rho) := \sum_{h=0}^\infty \gamma^h \mathbb{E}_{s_0 \sim \rho, a_h \sim \pi(a_h|s_h)} [r(s_h, a_h)] + \alpha \mathbb{H}(\rho, \pi) \tag{4}$$

and $\mathbb{H}(\pi(a_h|s_h))$ is the discounted entropy term

$$\mathbb{H}(\rho, \pi) := \mathbb{E}_{s_0 \sim \rho, a_h \sim \pi(\cdot|s_h)} \left[ \sum_{h=0}^\infty -\gamma^h \log \pi(a_h|s_h) \right], \tag{5}$$

and $\alpha$ is the penalty term. For simplicity, we denote the optimization objective function in (4) as $\alpha$-MaxEnt RL. Similar to Equation 18 & 19 of Mei et al. (2020), we also define the advantage and $Q$-functions and for MaxEnt RL as

$$A^\pi(s, a) := Q^\pi(s, a) - \alpha \log \pi(s, a) - V^\pi(s),$$
$$Q^\pi(s, a) := r(s, a) + \gamma \sum_{s'} P(s'|s, a) V^\pi(s). \tag{6}$$

2. We let

$$d_{s_0}^\pi(s) = (1 - \gamma) \sum_{t=0}^\infty \gamma^t \mathbb{P}^\pi(s_t = s|s_0), \tag{7}$$

to denote the discounted state visitation of policy $\pi$ starting at state $s_0$, and let

$$d_\rho^\pi(s) = \mathbb{E}_{s \sim \rho}[d_s^\pi(s)] \tag{8}$$

denote the initial state visitation distribution under **initial state distribution** $\rho$.

3. We assume the reward functions under all context are bounded within $[0, 1]$:

$$r_\omega(s, a) \in [0, 1], \ \forall \omega \in \Omega, \forall(s, a) \in \mathcal{S} \times \mathcal{A}. \tag{9}$$

4. Similar to previous analysis in Agarwal et al. (2021); Mei et al. (2020); Ding et al. (2021), we assume the initial distribution $\mu$ for PG/stochastic PG satisfies $\rho(s) > 0, \forall s \in \mathcal{S}$.

## A.2  MAIN RESULTS: MISMATCH COEFFICIENT UPPER BOUND

**Lemma A.1 (Density Mismatch Ratio via ROLLIN)** *Using* (1) *from* ROLLIN $\mu_k = \beta d_{\mu_{k-1}}^{\pi^\star_{\omega_{k-1}}} + (1 - \beta)\rho$, *the density mismatch ratio* $\left\| d_{\mu_k}^{\pi^\star_{\omega_k}} / \mu_k \right\|_\infty$ *satisfies*

$$\left\| \frac{d_{\mu_k}^{\pi^\star_{\omega_k}}}{\mu_k} \right\|_\infty \leq \widetilde{\Omega} \left( \frac{L_r}{\alpha(1 - \beta)} \Delta_\omega^k S \right), \tag{10}$$

*where* $\Delta_\omega^k = \max_{1 \leq i \leq k} \|\omega_i - \omega_{i-1}\|_2$.

**Proof** By (1) from ROLLIN, we have

$$\left\| \frac{d_{\mu_k}^{\pi^\star_{\omega_k}}}{\mu_k} \right\|_\infty = \left\| \frac{d_{\mu_k}^{\pi^\star_{\omega_k}} - d_{\mu_{k-1}}^{\pi^\star_{\omega_{k-1}}} + d_{\mu_{k-1}}^{\pi^\star_{\omega_{k-1}}}}{\mu_k} \right\|_\infty$$

$$\overset{(i)}{\leq} \frac{\left\| d_{\mu_k}^{\pi^\star_{\omega_k}} - d_{\mu_{k-1}}^{\pi^\star_{\omega_{k-1}}} \right\|_1}{\min \mu_k} + \left\| \frac{d_{\mu_{k-1}}^{\pi^\star_{\omega_{k-1}}}}{\beta d_{\mu_{k-1}}^{\pi^\star_{\omega_{k-1}}} + (1 - \beta)\rho} \right\|_\infty \tag{11}$$

$$\overset{(ii)}{\leq} \frac{\left\| d_{\mu_k}^{\pi^\star_{\omega_k}} - d_{\mu_{k-1}}^{\pi^\star_{\omega_{k-1}}} \right\|_1}{\min \mu_k} + \frac{1}{\beta}$$

where inequality $(i)$ holds because of (1), and inequality $(ii)$ holds by setting $\rho = 0$. Now it remains to bound $\left\| d_{\mu_{k+1}}^{\pi_{\omega_{k+1}}^\star} - d_{\mu_k}^{\pi_{\omega_k}^\star} \right\|_1$ using the difference $\|\omega_{k+1} - \omega_k\|_2$. Let $\mathbb{P}_h^k = \mathbb{P}_h^{\pi_{\omega_k}^\star}(s'|s_0 \sim \mu_k)$ denote the state visitation distribution resulting from $\pi_{\omega_k}^\star$ probability starting at $\mu_k$, then we have

$$
\mathbb{P}_h^k(s') - \mathbb{P}_h^{k-1}(s') = \sum_{s,a} \left( \mathbb{P}_{h-1}^k(s)\pi_{\omega_k}^\star(a|s) - \mathbb{P}_{h-1}^{k-1}(s)\pi_{\omega_{k-1}}^\star(a|s) \right) P(s'|s,a)
$$

$$
= \sum_{s,a} \left( \mathbb{P}_{h-1}^k(s)\pi_{\omega_k}^\star(a|s) - \mathbb{P}_{h-1}^k(s)\pi_{\omega_{k-1}}^\star(a|s) + \mathbb{P}_{h-1}^{k-1}(s)\pi_{\omega_{k-1}}^\star(a|s) - \mathbb{P}_{h-1}^{k-1}(s)\pi_{\omega_{k-1}}^\star(a|s) \right) P(s'|s,a)
$$

$$
= \sum_{s} \mathbb{P}_{h-1}^k(s) \left[ \sum_{a} \left( \pi_{\omega_k}^\star(a|s) - \pi_{\omega_{k-1}}^\star(a|s) \right) P(s'|s,a) \right]
$$

$$
+ \sum_{s} \left( \mathbb{P}_{h-1}^k(s) - \mathbb{P}_{h-1}^{k-1}(s) \right) \left[ \sum_{a} \pi_{\omega_{k-1}}^\star(a|s) P(s'|s,a) \right].
$$

$$(12)$$

Taking absolute value on both side, yields

$$
\left\| \mathbb{P}_h^k - \mathbb{P}_h^{k-1} \right\|_1 = \sum_{s'} \left| \mathbb{P}_h^k(s') - \mathbb{P}_h^{k-1}(s') \right|
$$

$$
\leq \sum_{s} \mathbb{P}_{h-1}^k(s) \sum_{a} \underbrace{\left| \pi_{\omega_k}^\star(a|s) - \pi_{\omega_{k-1}}^\star(a|s) \right|}_{\leq c_1 \|\omega_k - \omega_{k-1}\|_2} \sum_{s'} P(s'|s,a)
$$

$$
+ \sum_{s} \left| \mathbb{P}_{h-1}^k(s) - \mathbb{P}_{h-1}^{k-1}(s) \right| \left[ \sum_{s'} \sum_{a} \pi_{\omega_{k-1}}^\star(a|s) P(s'|s,a) \right]
$$

$$
\overset{(i)}{\leq} c_1 \|\omega_k - \omega_{k-1}\|_2 + \left\| \mathbb{P}_{h-1}^k - \mathbb{P}_{h-1}^{k-1} \right\|_1 \leq \cdots \leq c_1 h \|\omega_k - \omega_{k-1}\|_2 + \left\| \mathbb{P}_0^k - \mathbb{P}_0^{k-1} \right\|_1
$$

$$
\overset{(ii)}{=} c_1 h \|\omega_k - \omega_{k-1}\|_2 + \|\mu_k - \mu_{k-1}\|_1,
$$

$$(13)$$

where inequality $(i)$ holds by applying Lemma B.2 with $c_1 = L_r/\alpha(1-\gamma)$ from and equality $(ii)$ holds because the initial distribution of $\mathbb{P}_h^k$ is $\mu_k$. By the definition of $d_\mu^\pi$, we have

$$
d_{\mu_k}^{\pi_{\omega_k}^\star}(s) - d_{\mu_{k-1}}^{\pi_{\omega_{k-1}}^\star}(s) \overset{(i)}{=} d_k(s) - d_{k-1}(s) = (1-\gamma) \sum_{h=0}^{\infty} \gamma^h \left( \mathbb{P}_h^k(s) - \mathbb{P}_h^{k-1}(s) \right), \; \forall s \in \mathcal{S}. \quad (14)
$$

where in equality $(i)$, we use $d_k$ to denote $d_{\mu_k}^{\pi_{\omega_k}^\star}$. Adding $\ell^1$ norm on both sides of (14) and applying (13), yields

$$
\|d_k - d_{k-1}\|_1 \leq (1-\gamma) \sum_{h=0}^{\infty} \gamma^h \left( c_1 h \|\omega_k - \omega_{k-1}\|_2 + \|\mu_k - \mu_{k-1}\|_1 \right)
$$

$$
\overset{(i)}{=} \frac{\gamma c_1}{1-\gamma} \|\omega_k - \omega_{k-1}\|_2 + \|\mu_k - \mu_{k-1}\|_1 \overset{(ii)}{=} \frac{\gamma c_1}{1-\gamma} \|\omega_k - \omega_{k-1}\|_2 + \beta \|d_{k-1} - d_{k-2}\|_1,
$$

$$(15)$$

where equality $(i)$ holds because $\sum_{h=0}^{\infty} \gamma^h h = \gamma/(1-\gamma)^2$ and equality $(ii)$ holds because of (1). Hence, we know that

$$
\|d_k - d_{k-1}\|_1 \leq \frac{\gamma c_1}{1-\gamma} \|\omega_k - \omega_{k-1}\|_2 + \beta \|d_{k-1} - d_{k-2}\|_1
$$

$$
\leq \frac{\gamma c_1}{1-\gamma} \sum_{i=0}^{k-1} \left[ \|\omega_{i+1} - \omega_i\|_2 \beta^{k-i} \right] + \beta^{k-1} \|d_1 - d_0\|_1 \quad (16)
$$

$$
\leq \frac{\gamma c_1}{1-\gamma} \cdot \frac{1}{1-\beta} \Delta_\omega^k + \beta^{k-1} \|d_1 - d_0\|_1 \approx \frac{\gamma c_1}{(1-\gamma)(1-\beta)} \Delta_\omega^k
$$

where $\Delta_\omega^k = \max_{1 \le i \le k} \|\omega_i - \omega_{i-1}\|_2$ and the last $\approx$ holds when $k$ is large. Therefore, applying (16) back to (11), we know that

$$
\begin{aligned}
\left\| \frac{d_{\mu_k}^{\pi_{\omega_k}^\star}}{\mu_k} \right\|_\infty &\le \frac{\left\| d_{\mu_k}^{\pi_{\omega_k}^\star} - d_{\mu_{k-1}}^{\pi_{\omega_{k-1}}^\star} \right\|_1}{\min \mu_k} + \frac{1}{\beta} \\
&\overset{(i)}{\le} \frac{1}{\min \mu_k} \frac{\gamma c_1}{(1-\gamma)(1-\beta)} \Delta_\omega^k + \frac{1}{\beta} = \widetilde{\Omega}\left( \frac{L_r}{\alpha(1-\beta)} \Delta_\omega^k S \right),
\end{aligned}
\tag{17}
$$

where inequality $(i)$ holds since Lemma B.2 implies $c_1 = L_r/\alpha(1-\gamma)$, and we omit the $1/(1-\gamma)^6$ and $\log$ in the $\widetilde{\Omega}$, which completes the proof. Note that we can only achieve the final bound $\widetilde{\Omega}\left( \frac{L_r}{\alpha(1-\beta)} \Delta_\omega^k S \right)$ by setting $\beta$ as a constant. If we pick an arbitrarily small $\beta$, then the $1/\beta$ term will dominate the complexity and we will not have the final bound of $\widetilde{\Omega}\left( \frac{L_r}{\alpha(1-\beta)} \Delta_\omega^k S \right)$. $\blacksquare$

### A.3 COMPLEXITY OF VANILLA STOCHASTIC PG

**Theorem A.2 (Complexity of Stochastic PG (Theorem 5.1 of Ding et al. (2021)))** *Consider an arbitrary tolerance level $\delta > 0$ and a small enough tolerance level $\varepsilon > 0$. For every initial point $\theta_1$, if $\theta_{T+1}$ is generated by SPG (Algorithm 4) with*

$$
\begin{aligned}
&T_1 \ge \left( \frac{6D(\theta_0)}{\delta \varepsilon_0} \right)^{\frac{8L}{C_\delta^0 \ln 2}}, \ T_2 \ge \left( \frac{\varepsilon_0}{6\delta\varepsilon} - 1 \right) t_0, \ T = T_1 + T_2, \\
&B_1 \ge \max\left\{ \frac{30\sigma^2}{C_\delta^0 \varepsilon_0 \delta}, \frac{6\sigma T_1 \log T_1}{\bar{\Delta} L} \right\}, \ B_2 \ge \frac{\sigma^2 \ln(T_2 + t_0)}{6C_\zeta \delta\varepsilon}, \\
&\eta_t = \eta \le \min\left\{ \frac{\log T_1}{T_1 L}, \frac{8}{C_\delta^0}, \frac{1}{2L} \right\} \ \forall 1 \le t \le T_1, \ \eta_t = \frac{1}{t - T_1 + t_0} \ \forall t > T_1,
\end{aligned}
\tag{18}
$$

*where*

$$
\begin{aligned}
&D(\theta_t) = V^{\pi^\star}(\rho) - V^{\pi_{\theta_t}}(\rho), \ \varepsilon_0 = \min\left\{ \left( \frac{\alpha \min_{s \in \mathcal{S}} \rho(s)}{6 \ln 2} \right)^2 \left[ \zeta \exp\left( -\frac{1}{(1-\gamma)\alpha} \right) \right]^4, 1 \right\}, \\
&t_0 \ge \sqrt{\frac{3\sigma^2}{2\delta\varepsilon_0}}, \ C_\delta^0 = \frac{2\alpha}{S} \left\| \frac{d_\rho^{\pi^\star}}{\rho} \right\|_\infty^{-1} \min_{s \in \mathcal{S}} \rho(s) \min_{\theta \in \mathcal{G}_\delta^0} \min_{s,a} \pi_\theta(a|s)^2, \\
&C_\zeta = \frac{2\alpha}{S} \left\| \frac{d_\rho^{\pi^\star}}{\rho} \right\|_\infty^{-1} \min_{s \in \mathcal{S}} \rho(s)(1-\zeta)^2 \min_{s,a} \pi^\star(a|s)^2, \\
&\mathcal{G}_\delta^0 := \left\{ \theta \in \mathbb{R}^{S \times A} : \min_{\theta^\star \in \Theta^\star} \|\theta - \theta^\star\|_2 \le (1 + 1/\delta)\bar{\Delta} \right\}, \ \bar{\Delta} = \left\| \log c_{\bar{\theta}_1, \eta} - \log \pi^\star \right\|_2, \\
&c_{\bar{\theta}_1, \eta} = \inf_{t \ge 1} \min_{s,a} \pi_{\theta_t}(a|s), \ \sigma^2 = \frac{8}{(1-\gamma)^2} \left( \frac{1 + (\alpha \log A)^2}{(1-\gamma^{1/2})^2} \right), \ L = \frac{8 + \alpha(4 + 8\log A)}{(1-\gamma)^3},
\end{aligned}
\tag{19}
$$

*then we have $\mathbb{P}(D(\theta_{T+1} \le \varepsilon)) \ge 1 - \delta$.*

**Corollary A.3 (Iteration Complexity and Sample Complexity for $\varepsilon$-Optimal Policies)** *Suppose we set the tolerance level $\varepsilon, \delta = O(S^{-1})$, the iteration complexity and sample complexity of obtaining an $\varepsilon$-optimal policy using stochastic softmax policy gradient (Algorithm 4) in phase 1 and phase 2 satisfies:*

- *Phase 1: $T_1 = \widetilde{\Omega}\left( S^{2S^3} \right), B_1 = \widetilde{\Omega}\left( S^{2S^3} \right),$*

- *Phase 2: $T_2 = \widetilde{\Omega}\left( S^{3/2} \right), B_2 = \widetilde{\Omega}\left( S^5 \right),$*

*with probability at least $1 - \delta$.*

**Proof** We first check the dependency of (19) on $S$. Notice that

- $\varepsilon_0$:

$$\frac{1}{\varepsilon_0} = \max\left\{\left(\frac{6\ln 2}{\alpha \min_{s\in\mathcal{S}} \rho(s)}\right)^2 \left[\zeta \exp\left(-\frac{1}{(1-\gamma)\alpha}\right)\right]^{-4}, 1\right\} = \widetilde{\Omega}(S^2); \quad (20)$$

- $t_0$:

$$t_0 \geq \sqrt{\frac{3\sigma^2}{2\delta\varepsilon_0}} = \widetilde{\Omega}(S); \quad (21)$$

- $C_\delta^0$:

$$\frac{1}{C_\delta^0} = \frac{S}{2\alpha}\left\|\frac{d_\rho^{\pi^\star}}{\rho}\right\|_\infty \max_{s\in\mathcal{S}}\rho(s)^{-1}\frac{1}{\min_{\theta\in\mathcal{G}_\delta^0}\min_{s,a}\pi_\theta(a|s)^2} = \widetilde{\Omega}(S^3); \quad (22)$$

- $C_\zeta$:

$$\frac{1}{C_\zeta} = \frac{S}{2\alpha}\left\|\frac{d_\rho^{\pi^\star}}{\rho}\right\|_\infty \max_{s\in\mathcal{S}}\rho(s)^{-1}(1-\zeta)^{-2}\max_{s,a}\pi^\star(a|s)^{-2} = \widetilde{\Omega}(S^3). \quad (23)$$

Hence, the complexities in phase 1 scales at

$$T_1 \geq \left(\frac{6D(\theta_0)}{\delta\varepsilon_0}\right)^{\frac{8L}{C_\delta^0\ln 2}} = \widetilde{\Omega}\left(S^{2S^3}\right), \ B_1 \geq \max\left\{\frac{30\sigma^2}{C_\delta^0\varepsilon_0\delta}, \frac{6\sigma T_1\log T_1}{\bar{\Delta}L}\right\} = \widetilde{\Omega}\left(S^{2S^3}\right). \quad (24)$$

To enforce a positive $T_2$, the tolerance level $\varepsilon, \delta$ should satisfy $\frac{\varepsilon_0}{6\delta\varepsilon} \geq 1$, which implies $\frac{1}{\delta\varepsilon} = \Omega(S^2)$. Hence, if assuming $\frac{\varepsilon_0}{\delta\varepsilon} = o(S)$ the tolerance level $\varepsilon, \delta = O(S^{-1})$, the complexities in phase 2 scales at

$$T_2 \geq \left(\frac{\varepsilon_0}{6\delta\varepsilon} - 1\right)t_0 = \widetilde{\Omega}\left(S^{3/2}\right), \ B_2 \geq \frac{\sigma^2\ln(T_2 + t_0)}{6C_\zeta\delta\varepsilon} = \widetilde{\Omega}\left(S^5\right). \quad (25)$$

∎

### A.4  COMPLEXITY OF LEARNING THE NEXT CONTEXT

**Theorem A.4 (Theorem 4.1: Complexity of Learning the Next Context)** *Consider the context-based stochastic softmax policy gradient (line 7 of Algorithm 1), suppose Assumption 3.1 and Assumption 3.2 hold, then the iteration number of obtaining an $\varepsilon$-optimal policy for $\omega_k$ from $\theta_{\omega_k-1}^\star$ is $\widetilde{\Omega}\left(S^{3/2}\right)$ and the per iteration sample complexity is $\widetilde{\Omega}\left(\frac{L_r}{\alpha(1-\beta)}S^3\right)$.*

We first introduce the following lemma to aid the proof of Theorem A.4.

**Lemma A.5 (Bounded Optimal Values Between two Adjacent Contexts)** *Under the same conditions as Theorem A.4, we have*

$$V_{\omega_k}^{\pi_{\omega_k}^\star}(\rho) - V_{\omega_k}^{\pi_{\omega_{k-1}}^\star}(\rho) \leq \frac{2L_r\|\omega_k - \omega_{k-1}\|_2}{(1-\gamma)^2}. \quad (26)$$

**Proof** Let $V_\omega^\pi$ denote the value function of policy $\pi$ with reward function $r_\omega$. From (65) of Lemma B.3, we know that for any initial distribution $\rho$, we have

$$V_{\omega_k}^{\pi_{\omega_k}^\star}(\rho) - V_{\omega_k}^{\pi_{\omega_{k-1}}^\star}(\rho) = \frac{1}{1-\gamma}\sum_s\left[d_\rho^{\pi_{\omega_{k-1}}^\star}(s)\cdot\alpha\cdot D_{\mathrm{KL}}\left(\pi_{\omega_{k-1}}^\star(\cdot|s)\|\pi_{\omega_k}^\star(\cdot|s)\right)\right]. \quad (27)$$

From (47) of Lemma B.1, we know that

$$\pi^\star_{\omega_{k-1}}(a|s) = \left[\text{softmax}(Q^{\pi^\star_{\omega_{k-1}}}(\cdot, s)/\alpha)\right]_a := \frac{\exp\left[Q^{\pi^\star_{\omega_{k-1}}}(s, a)/\alpha\right]}{\sum_{a'} \exp\left[Q^{\pi^\star_{\omega_{k-1}}}(s, a')/\alpha\right]}$$

$$\pi^\star_{\omega_k}(a|s) = \left[\text{softmax}(Q^{\pi^\star_\omega}(\cdot, s)/\alpha)\right]_a := \frac{\exp\left[Q^{\pi^\star_{\omega_k}}(s, a)/\alpha\right]}{\sum_{a'} \exp\left[Q^{\pi^\star_{\omega_k}}(s, a')/\alpha\right]},$$
(28)

hence, we have

$$D_{\text{KL}}\left(\pi^\star_{\omega_{k-1}}(\cdot|s) || \pi^\star_{\omega_k}(\cdot|s)\right)$$
$$= \sum_a \pi^\star_{\omega_{k-1}}(a|s) \left\{ \log\left(\left[\text{softmax}(Q^{\pi^\star_{\omega_{k-1}}}(a, s)/\alpha)\right]_a\right) - \log\left(\left[\text{softmax}(Q^{\pi^\star_{\omega_k}}(a, s)/\alpha)\right]_a\right) \right\}.$$
(29)

Let $f(\boldsymbol{x})$ denote the log soft max function for an input vector $\boldsymbol{x} = [x_1, x_2, \ldots, x_A]^\top$ such that $x_i \geq 0$, then for a small perturbation $\boldsymbol{\Delta} \in \mathbb{R}^A$, the intermediate value theorem implies

$$|[f(\boldsymbol{x} + \boldsymbol{\Delta})]_i - [f(\boldsymbol{x})]_i| = \left|\boldsymbol{\Delta}^\top \nabla_{\boldsymbol{z}}[f(\boldsymbol{z})]_i\right|,$$
(30)

for some vector $\boldsymbol{z}$ on the segment $[\boldsymbol{x}, \boldsymbol{x} + \boldsymbol{\Delta}]$. Now consider the Jacobian of the log softmax function $\partial[\nabla_{\boldsymbol{z}} f(\boldsymbol{z})]_i / \partial z_j$:

$$\frac{\partial[\nabla_{\boldsymbol{z}} f(\boldsymbol{z})]_i}{\partial z_j} = \begin{cases} 1 - p_i(\boldsymbol{z}) \in (0, 1) & \text{if } i = j, \\ -p_j(\boldsymbol{z}) \in (-1, 0) & \text{otherwise}, \end{cases}$$
(31)

where $p_i(\boldsymbol{z}) = \exp(z_i) / \sum_{k=1}^A \exp(z_k)$. hence, we know that

$$|[f(\boldsymbol{x} + \boldsymbol{\Delta})]_i - [f(\boldsymbol{x})]_i| = \left|\boldsymbol{\Delta}^\top \nabla_{\boldsymbol{z}}[f(\boldsymbol{z})]_i\right| \leq \|\boldsymbol{\Delta}\|_\infty \sum_{k=1}^A \left|\frac{\partial[f(\boldsymbol{z})]_i}{\partial z_k}\right|$$

$$= \|\boldsymbol{\Delta}\|_\infty \left(1 - p_i(\boldsymbol{z}) + \sum_{j \neq i} p_j(\boldsymbol{z})\right) \leq 2\|\boldsymbol{\Delta}\|_\infty.$$
(32)

Now let

$$\boldsymbol{x} = \frac{1}{\alpha}[Q^{\pi^\star_{\omega_{k-1}}}(s, a_1), Q^{\pi^\star_{\omega_{k-1}}}(s, a_2), \ldots, Q^{\pi^\star_{\omega_{k-1}}}(s, a_A)],$$

$$\boldsymbol{x} + \boldsymbol{\Delta} = \frac{1}{\alpha}[Q^{\pi^\star_{\omega_k}}(s, a_1), Q^{\pi^\star_{\omega_k}}(s, a_2), \ldots, Q^{\pi^\star_{\omega_k}}(s, a_A)],$$
(33)

(57) from Lemma B.2 implies that

$$\frac{1}{\alpha}\left\|Q^{\pi^\star_{\omega_k}} - Q^{\pi^\star_{\omega_{k-1}}}\right\|_\infty \leq \frac{L_r \|\omega_k - \omega_{k-1}\|_2}{\alpha(1 - \gamma)},$$
(34)

substituting (34) and (32) into (29), yields

$$D_{\text{KL}}\left(\pi^\star_{\omega_{k-1}}(\cdot|s) || \pi^\star_{\omega_k}(\cdot|s)\right) \leq \sum_a 2\pi^\star_{\omega_{k-1}}(a|s)\|\boldsymbol{\Delta}\|_\infty \leq 2\|\boldsymbol{\Delta}\|_\infty \leq \frac{2L_r \|\omega_k - \omega_{k-1}\|_2}{\alpha(1 - \gamma)}.$$
(35)

Combine (35) with (27), we have

$$V^{\pi^\star_{\omega_k}}_{\omega_k}(\rho) - V^{\pi^\star_{\omega_{k-1}}}_{\omega_k}(\rho) = \frac{1}{1 - \gamma} \sum_s \left[d^{\pi^\star_{\omega_{k-1}}}_\rho(s) \cdot \alpha \cdot D_{\text{KL}}\left(\pi^\star_{\omega_{k-1}}(\cdot|s) || \pi^\star_{\omega_k}(\cdot|s)\right)\right]$$

$$\leq \frac{2L_r \|\omega_k - \omega_{k-1}\|_2}{(1 - \gamma)^2},$$
(36)

which completes the proof. ∎

Now we are ready to proceed to the proof of Theorem A.4.

**Proof** From (19) we know that

$$\varepsilon_0 = \min\left\{\left(\frac{\alpha \min_{s\in\mathcal{S}}\rho(s)}{6\ln 2}\right)^2\left[\zeta\exp\left(-\frac{1}{(1-\gamma)\alpha}\right)\right]^4, 1\right\} = O\left(\frac{1}{S^2}\right). \tag{37}$$

And from Section 6.2 of Ding et al. (2021), we can directly enter phase 2 of the stochastic PG when

$$V_{\omega_k}^{\pi_{\omega_k}^\star}(\rho) - V_{\omega_k}^{\pi_{\omega_{k-1}}^\star}(\rho) \le \varepsilon_0. \tag{38}$$

Hence, when $\Delta_\omega^k = \max_{1\le i\le k}\|\omega_i - \omega_{i-1}\|_2 = O(1/S^2)$, we have

$$V_{\omega_k}^{\pi_{\omega_k}^\star}(\rho) - V_{\omega_k}^{\pi_{\omega_{k-1}}^\star}(\rho) \le \frac{2L_r\Delta_\omega}{(1-\gamma)^2} \le \frac{\varepsilon_0}{2}, \tag{39}$$

which implies we can directly enter phase 2 and enjoys the faster iteration complexity of $T_2 = \Omega\left(S^{3/2}\right)$ (by choosing $\delta = O(S^{-1})$) and the smaller batch size of

$$B_2 \ge \frac{\sigma^2\ln(T_2+t_0)}{6C_\zeta\delta\varepsilon} \stackrel{(i)}{=} \widetilde{\Omega}\left(\frac{L_r}{\alpha(1-\beta)}\Delta_\omega^k S^5\right) \stackrel{(ii)}{=} \widetilde{\Omega}\left(\frac{L_r}{\alpha(1-\beta)}S^3\right), \tag{40}$$

where equation $(i)$ holds by applying Lemma A.1 to (23):

$$\frac{\sigma^2\ln(T_2+t_0)}{6C_\zeta\delta\varepsilon} = \widetilde{\Omega}\left(S^4\cdot\left\|d_{\mu_k}^{\pi_{\omega_k}^\star}/\mu_k\right\|_\infty\right) = \widetilde{\Omega}\left(\frac{L_r}{\alpha(1-\beta)}\Delta_\omega^k S^5\right),$$

and equality $(ii)$ holds by the assumption that $\Delta_\omega^k = O(S^{-2})$ and we omit the log term and components not related to $S$ in $\widetilde{\Omega}$. ∎

## A.5 TOTAL COMPLEXITY OF ROLLIN

**Theorem A.6 (Theorem 4.3: Total Complexity of Learning the Target Context)** *Suppose Assumption 3.1 and 3.2 hold, and $\theta_0^{(0)}$ is an near-optimal initialization, then the total number of iteration of learning $\pi_{\omega_K}^\star$ using Algorithm 1 is $\Omega(KS^{3/2})$ and the per iteration is $\widetilde{\Omega}\left(S^3\right)$, with high probability.*

**Proof** From lemma A.5, we know that

$$V_{\omega_k}^{\pi_{\omega_k}^\star}(\rho) - V_{\omega_k}^{\pi_{\omega_{k-1}}^\star}(\rho) \le \frac{2L_r\|\omega_k - \omega_{k-1}\|_2}{(1-\gamma)^2}. \tag{41}$$

Suppose for each context $\omega_k$, we initialize the parameters of the policy as $\theta_1^{(k)} = \theta_{\omega_{k-1}}^\star$, and let $\theta_t^{(k)}$ denote the parameters at the $t^{\text{th}}$ iteration of SPG. We will use induction to show that when $t = \widetilde{\Omega}(S^{3/2})$, $\forall k\in[K]$, we have

$$V_{\omega_k}^{\pi_{\omega_k}^\star}(\rho) - V_{\omega_k}^{\pi_{\theta_t^{(k-1)}}}(\rho) < \varepsilon_0, \tag{42}$$

this implies that for any context $\omega_k, k\in[K]$, we can always find a good initialization by setting $\theta_1^{(k)} = \theta_t^{(k-1)}$ from learning $\pi_{\omega_{k-1}}^\star$ using SPG after $t = \Omega(S^{3/2})$ iteration. This result guarantees that every initialization $\theta_1^{(k)}$ for learning the optimal contextual policy $\pi_{\omega_k}^\star$ will directly start from the efficient phase 2.

**Induction:** $k = 0$. When $k = 0$, Assumption 3.2 and the near-optimal initialization (Definition 4.2) of $\theta_0^{(0)}$ implies that

$$V_{\omega_0}^{\pi_{\omega_0}^\star}(\rho) - V_{\omega_0}^{\pi_{\theta_0^{(0)}}}(\rho) < \varepsilon_0. \tag{43}$$

This result implies that a near-optimal initialization allows the initialization to directly start from phase 2 of SPG.

**Induction: from $k-1$ to $k$.** Suppose the result in (42) holds for $k-1$, then we know that

$$V_{\omega_{k-1}}^{\pi^\star_{\omega_{k-1}}}(\rho) - V_{\omega_{k-1}}^{\pi_{\theta_1^{(k-1)}}}(\rho) = V_{\omega_{k-1}}^{\pi^\star_{\omega_{k-1}}}(\rho) - V_{\omega_{k-1}}^{\pi_{\theta_t^{(k-2)}}}(\rho) < \varepsilon_0. \tag{44}$$

Select $\varepsilon$ such that $\varepsilon \leq \varepsilon_0/2$. Theorem A.4 suggests that when $t' = \tilde{\Omega}(S^{3/2})$, with high probability, we have

$$V_{\omega_k}^{\pi^\star_{\omega_k}}(\rho) - V_{\omega_k}^{\pi_{\theta_{t'}^{(k-1)}}}(\rho) < \varepsilon \leq \frac{\varepsilon_0}{2}. \tag{45}$$

Hence, if we initialize $\theta_1^{(k)} = \theta_t^{(k-1)}$, with high probability when $t' = \tilde{\Omega}(S^{3/2})$, we have

$$
\begin{aligned}
& V_{\omega_k}^{\pi^\star_{\omega_k}}(\rho) - V_{\omega_k}^{\pi_{\theta_{t'}^{(k-1)}}}(\rho) = V_{\omega_k}^{\pi^\star_{\omega_k}}(\rho) - V_{\omega_k}^{\pi^\star_{\omega_{k-1}}}(\rho) + V_{\omega_k}^{\pi^\star_{\omega_{k-1}}}(\rho) - V_{\omega_k}^{\pi_{\theta_{t'}^{(k-1)}}}(\rho) \\
& \overset{(i)}{\leq} \frac{\varepsilon_0}{2} + V_{\omega_k}^{\pi^\star_{\omega_k}}(\rho) - V_{\omega_k}^{\pi_{\theta_{t'}^{(k-1)}}}(\rho) \overset{(ii)}{<} \varepsilon_0,
\end{aligned}
\tag{46}
$$

where inequality $(i)$ holds by equation (39) in Theorem A.4, inequality $(ii)$ holds because of the induction assumption in (45).

Therefore, we have shown (42) holds for $t = \tilde{\Omega}(S^{3/2}), \forall k \in [K]$. Since we have $K$ contexts in total, we know that Algorithm 1 can enforce a good initialization $\theta_1^{(k)}$ that directly starts from phase 2 for learning all $\pi^\star_{\omega_k}$, and for each $k \in [K]$, the iteration complexity is $\tilde{\Omega}(S^{3/2})$. Hence the total iteration complexity of obtaining an $\varepsilon$-optimal policy for the final context $\omega_K$ is $\tilde{\Omega}(KS^{3/2})$, with per iteration sample complexity of $\tilde{\Omega}(S^3)$. ∎

## APPENDIX B    KEY LEMMAS

### B.1    OPTIMAL POLICY OF MAXIMUM ENTROPY RL NACHUM ET AL. (2017)

**Lemma B.1** *The optimal policy $\pi^\star$ that maximizes the $\alpha$-MaxEnt RL objective (4) with penalty term $\alpha$ satisfies:*

$$\pi^\star(a|s) = \exp\left[\left(Q^{\pi^\star}(s,a) - V^{\pi^\star}(s)\right)/\alpha\right] = \frac{\exp\left(Q^{\pi^\star}(s,a)/\alpha\right)}{\sum_a \exp\left(Q^{\pi^\star}(s,a)/\alpha\right)} \tag{47}$$

*for all $h \in \mathbb{N}$, where*

$$
\begin{aligned}
Q^{\pi^\star}(s,a) &:= r(s,a) + \gamma \mathbb{E}_{s' \sim P(s'|s,a)} V(s') \\
V^{\pi^\star}(s) &:= \alpha \log\left(\sum_a \exp\left(Q^{\pi^\star}(s,a)/\alpha\right)\right).
\end{aligned}
\tag{48}
$$

**Proof** Similar proof appears in (Nachum et al., 2017), we provide the proof for completeness. At the optimal policy $\pi_\theta = \pi^\star$, take the gradient of (4) w.r.t. $p \in \Delta(\mathcal{A})$ and set it to 0, we have

$$\frac{\partial}{\partial p(a)}\left[\sum_{a \in \mathcal{A}} p(a)\left(Q^{\pi^\star}(s,a) - \alpha \ln p(a)\right)\right] = Q^{\pi^\star}(s,a) - \alpha \ln p(a) - \alpha = 0, \tag{49}$$

which implies

$$p(a) = \exp\left(\frac{Q^{\pi^\star}(s,a)}{\alpha} - 1\right) \propto \exp\left(\frac{Q^{\pi^\star}(s,a)}{\alpha}\right). \tag{50}$$

Hence, we conclude that $\pi^\star(a|s) \propto \exp(Q^\star(s,a)/\alpha)$. ∎

### B.2    BOUNDING THE DIFFERENCE BETWEEN OPTIMAL POLICIES

**Lemma B.2** *Suppose Assumption 3.1 holds, let $\pi^\star_\omega(a|s), \pi^\star_{\omega'}(a|s)$ denote the optimal policy for $\alpha$-MaxEnt RL (47), then $\forall (s,a) \in \mathcal{S} \times \mathcal{A}$, the optimal policies of $\alpha$-MaxEnt RL under context $\omega, \omega'$ satisfy:*

$$|\pi^\star_\omega(a|s) - \pi^\star_{\omega'}(a|s)| \leq \frac{L_r \|\omega - \omega'\|_2}{\alpha(1-\gamma)}. \tag{51}$$

**Proof** From Lemma C.1, we know that the soft value iteration

$$\mathcal{T}Q(s,a) = r(s,a) + \gamma\alpha\mathbb{E}_{s'}\left[\log\sum_{a'}\exp Q(s',a')/\alpha\right] \tag{52}$$

is a contraction. Let $Q_\omega^t, Q_{\omega'}^t$ denote the Q functions at the $t^{\text{th}}$ value iteration under context $\omega, \omega'$ respectively, we know $Q_\omega^\infty = Q_\omega^{\pi^\star}$ and $Q_{\omega'}^\infty = Q_{\omega'}^{\pi^\star}$. Let $\varepsilon_t = \|Q_\omega^t - Q_{\omega'}^t\|_\infty$, then we have

$$\varepsilon_{t+1} = \|Q_\omega^{t+1} - Q_{\omega'}^{t+1}\|_\infty$$

$$= \left\|r_\omega(s,a) - r_{\omega'}(s,a) + \gamma\alpha\mathbb{E}_{s'}\left[\log\sum_{a'}\exp\frac{Q_{\omega'}^t(s',a')}{\alpha}\right] - \gamma\alpha\mathbb{E}_{s'}\left[\log\sum_{a'}\exp\frac{Q_{\omega'}^t(s',a')}{\alpha}\right]\right\|_\infty$$

$$\leq \|r_\omega - r_{\omega'}\|_\infty + \gamma\alpha\left\|\mathbb{E}_{s'}\log\sum_{s'}\exp Q_\omega^t(s',a')/\alpha - \mathbb{E}_{s'}\log\sum_{s'}\exp Q_{\omega'}^t(s',a')/\alpha\right\|_\infty$$

$$\leq \|r_\omega - r_{\omega'}\|_\infty + \gamma\|Q_\omega^t - Q_{\omega'}^t\|_\infty = \|r_\omega - r_{\omega'}\|_\infty + \gamma\varepsilon_t, \tag{53}$$

where the last inequality holds because $f(\boldsymbol{x}) = \log\sum_{i=1}^n \exp(x_i)$ is a contraction. From (53), we have

$$\varepsilon_{t+1} \leq \|r_\omega - r_{\omega'}\|_\infty + \gamma\varepsilon_t \leq (1+\gamma)\|r_\omega - r_{\omega'}\|_\infty + \gamma^2\varepsilon_{t-1} \leq \cdots \leq \|r_\omega - r_{\omega'}\|_\infty\sum_{i=0}^t\gamma^i + \gamma^t\varepsilon_1, \tag{54}$$

which implies

$$\left\|Q_\omega^{\pi^\star} - Q_{\omega'}^{\pi^\star}\right\|_\infty = \varepsilon_\infty \leq \frac{\|r_\omega - r_{\omega'}\|_\infty}{1-\gamma} \leq \frac{L_r\|\omega - \omega'\|_2}{1-\gamma}, \tag{55}$$

where the last inequality holds by Assumption 3.1. Hence, we have

$$\frac{1}{\alpha}\left|Q^{\pi_\omega^\star}(s,a) - Q^{\pi_{\omega'}^\star}(s,a)\right| \leq \frac{L_r\|\omega - \omega'\|_2}{\alpha(1-\gamma)}, \ \forall s,a \in \mathcal{S}\times\mathcal{A} \tag{56}$$

which implies

$$\frac{1}{\alpha}\left\|Q^{\pi_\omega^\star} - Q^{\pi_{\omega'}^\star}\right\|_\infty \leq \frac{L_r\|\omega - \omega'\|_2}{\alpha(1-\gamma)}. \tag{57}$$

Next, let $\pi_\omega^\star, \pi_{\omega'}^\star$ denote the maximum entropy policy RL under context $\omega, \omega'$ respectively. Then for a fixed state action pair $(s,a) \in \mathcal{S}\times\mathcal{A}$, we have

$$\pi_\omega^\star(a|s) = \left[\text{softmax}(Q^{\pi_\omega^\star}(\cdot,s)/\alpha)\right]_a := \frac{\exp\left[Q^{\pi_\omega^\star}(s,a)/\alpha\right]}{\sum_{a'}\exp\left[Q^{\pi_\omega^\star}(s,a')/\alpha\right]},$$
$$\pi_{\omega'}^\star(a|s) = \left[\text{softmax}(Q^{\pi_{\omega'}^\star}(\cdot,s)/\alpha)\right]_a := \frac{\exp\left[Q^{\pi_{\omega'}^\star}(s,a)/\alpha\right]}{\sum_{a'}\exp\left[Q^{\pi_{\omega'}^\star}(s,a')/\alpha\right]}, \tag{58}$$

where $Q^{\pi_\omega^\star}(\cdot,s), Q^{\pi_{\omega'}^\star}(\cdot,s) \in \mathbb{R}^A$, and we want to bound $|\pi_\omega^\star(a|s) - \pi_{\omega'}^\star(a|s)|$. Next we will use (57) to bound $|\pi_\omega^\star(a|s) - \pi_{\omega'}^\star(a|s)|$, where the last inequality holds by (56). Let $f(\boldsymbol{x})$ denote the softmax function for an input vector $\boldsymbol{x} = [x_1, x_2, \ldots, x_A]^\top$ such that $x_i \geq 0$, then for a small perturbation $\boldsymbol{\Delta} \in \mathbb{R}^A$, the intermediate value theorem implies

$$|[f(\boldsymbol{x} + \boldsymbol{\Delta})]_i - [f(\boldsymbol{x})]_i| = \left|\boldsymbol{\Delta}^\top\nabla_{\boldsymbol{x}}[f(\boldsymbol{z})]_i\right|, \tag{59}$$

for some vector $\boldsymbol{z}$ on the segment $[\boldsymbol{x}, \boldsymbol{x} + \boldsymbol{\Delta}]$. Hence

$$|[f(\boldsymbol{x} + \boldsymbol{\Delta})]_i - [f(\boldsymbol{x})]_i| = \left|\boldsymbol{\Delta}^\top[\nabla_{\boldsymbol{x}}f(\boldsymbol{z})]_i\right| \leq \|\boldsymbol{\Delta}\|_\infty\sum_{k=1}^A\left|\frac{\partial[f(\boldsymbol{z})]_i}{\partial z_k}\right|$$

$$\leq \|\boldsymbol{\Delta}\|_\infty\left(p_i(\boldsymbol{z})(1-p_i(\boldsymbol{z})) + \sum_{j\neq i}p_i(\boldsymbol{z})p_j(\boldsymbol{z})\right) < \|\boldsymbol{\Delta}\|_\infty\left(p_i(\boldsymbol{z}) + \sum_{j\neq i}p_j(\boldsymbol{z})\right) = \|\boldsymbol{\Delta}\|_\infty, \tag{60}$$

where the Jacobian of the softmax function $\partial \left[ \nabla_{\boldsymbol{x}} f(\boldsymbol{z}) \right]_i / \partial z_j$ satisfies:

$$\frac{\partial \left[ \nabla_{\boldsymbol{x}} f(\boldsymbol{z}) \right]_i}{\partial z_j} = \begin{cases} p_i(\boldsymbol{z})(1 - p_i(\boldsymbol{z})) & \text{if } i = j, \\ p_i(\boldsymbol{z}) p_j(\boldsymbol{z}) & \text{otherwise,} \end{cases} \tag{61}$$

and $p_i(\boldsymbol{z}) = \exp(z_i) / \sum_{k=1}^{A} \exp(z_k)$. Now let

$$\begin{aligned} \boldsymbol{x} &= \frac{1}{\alpha} [Q^{\pi_\omega^\star}(s, a_1), Q^{\pi_\omega^\star}(s, a_2), \dots, Q^{\pi_\omega^\star}(s, a_A)], \\ \boldsymbol{x} + \boldsymbol{\Delta} &= \frac{1}{\alpha} [Q^{\pi_{\omega'}^\star}(s, a_1), Q^{\pi_{\omega'}^\star}(s, a_2), \dots, Q^{\pi_{\omega'}^\star}(s, a_A)]. \end{aligned} \tag{62}$$

We know that $f(\boldsymbol{x}) = \pi_\omega^\star(a|s)$ and $f(\boldsymbol{x} + \boldsymbol{\Delta}) = \pi_{\omega'}^\star(a|s)$. Then (57) implies that

$$\|\boldsymbol{\Delta}\|_\infty \le \frac{L_r \|\omega - \omega'\|_2}{\alpha(1 - \gamma)}, \tag{63}$$

substituting this bound on $\|\boldsymbol{\Delta}\|_\infty$ into (60), we have

$$|\pi_\omega^\star(a|s) - \pi_{\omega'}^\star(a|s)| = |f(\boldsymbol{x}) - f(\boldsymbol{x} + \boldsymbol{\Delta})| \le \|\boldsymbol{\Delta}\|_\infty \le \frac{L_r \|\omega - \omega'\|_2}{\alpha(1 - \gamma)}, \tag{64}$$

which completes the proof. ∎

### B.3 SOFT SUB-OPTIMALITY LEMMA (LEMMA 25 & 26 OF MEI ET AL. (2020))

**Lemma B.3** *For any policy $\pi$ and any initial distribution $\rho$, the value function $V^\pi(\rho)$ of the $\alpha$-MaxEnt RL (48) satisfies:*

$$V^{\pi^\star}(\rho) - V^\pi(\rho) = \frac{1}{1 - \gamma} \sum_s \left[ d_\rho^\pi(s) \cdot \alpha \cdot D_{\mathrm{KL}}\left(\pi(\cdot|s) \| \pi^\star(\cdot|s)\right) \right], \tag{65}$$

*where $\pi^\star$ is the optimal policy of the $\alpha$-MaxEnt RL (4).*

**Proof** Similar proof appears in Lemma 25 & 26 of Mei et al. (2020), we provide the proof here for completeness.

**Soft performance difference.** We first show a soft performance difference result for the MaxEnt value function (Lemma 25 of Mei et al. (2020)). By the definition of MaxEnt value function and $Q$-function (4), (6), $\forall \pi, \pi'$, we have

$$\begin{aligned} &V^{\pi'}(s) - V^\pi(s) \\ =& \sum_a \pi'(a|s) \cdot \left[ Q^{\pi'}(s, a) - \alpha \log \pi'(a|s) \right] - \sum_a \pi(a|s) \cdot [Q^\pi(s, a) - \alpha \log \pi(a|s)] \\ =& \sum_a (\pi'(a|s) - \pi(a|s)) \cdot \left[ Q^{\pi'}(a|s) - \alpha \log \pi'(a|s) \right] \\ &+ \sum_a \pi(a|s) \cdot \left[ Q^{\pi'}(s, a) - \alpha \log \pi'(a|s) - Q^\pi(s, a) + \alpha \log \pi(a|s) \right] \\ =& \sum_a (\pi'(a|s) - \pi(a|s)) \cdot \left[ Q^{\pi'}(a|s) - \alpha \log \pi'(a|s) \right] + \alpha D_{\mathrm{KL}}\left(\pi(\cdot|s) \| \pi'(\cdot|s)\right) \\ &+ \gamma \sum_a \pi(a|s) \sum_{s'} P(s'|s, a) \cdot \left[ V^{\pi'}(s') - V^\pi(s') \right] \\ =& \frac{1}{1 - \gamma} \sum_{s'} d_s^\pi(s') \left[ \sum_{a'} (\pi'(a'|s') - \pi(a'|s')) \left[ Q^{\pi'}(s', a') - \alpha \log \pi'(a'|s') \right] \right. \\ &\left. + \alpha D_{\mathrm{KL}}\left(\pi(\cdot|s') \| \pi'(\cdot|s')\right) \right], \end{aligned} \tag{66}$$

where the last equality holds because by the definition of state visitation distribution

$$d_{s_0}^{\pi}(s) = (1-\gamma) \sum_{t=0}^{\infty} \gamma^t \mathbb{P}^{\pi}(s_t = s|s_0), \tag{67}$$

taking expectation of $s$ w.r.t. $s \sim \rho$, yields

$$
\begin{aligned}
&V^{\pi'}(\rho) - V^{\pi}(\rho) \\
&= \frac{1}{1-\gamma} \sum_{s'} d_{\rho}^{\pi}(s') \Bigg[ \sum_{a'} (\pi'(a'|s') - \pi(a'|s')) \cdot \Big[ Q^{\pi'}(s', a') - \alpha \log \pi'(a'|s') \Big] \\
&\quad + \alpha D_{\mathrm{KL}}\left(\pi(\cdot|s')||\pi'(\cdot|s')\right) \Bigg],
\end{aligned}
\tag{68}
$$

and (68) is known as the soft performance difference lemma (Lemma 25 in Mei et al. (2020)).

**Soft sub-optimality.** Next we will show the soft sub-optimality result. By the definition of the optimal policy of $\alpha$-MaxEnt RL (47), we have

$$\alpha \log \pi^{\star}(a|s) = Q^{\pi^{\star}}(s, a) - V^{\pi^{\star}}(s). \tag{69}$$

Substituting $\pi^{\star}$ into the performance difference lemma (68), we have

$$
\begin{aligned}
&V^{\pi^{\star}}(s) - V^{\pi}(s) \\
&= \frac{1}{1-\gamma} \sum_{s'} d_s^{\pi}(s') \cdot \Bigg[ \sum_{a'} (\pi^{\star}(a'|s') - \pi(a'|s')) \cdot \underbrace{\Big[ Q^{\pi^{\star}}(s', a') - \alpha \log \pi^{\star}(a'|s') \Big]}_{=V^{\pi^{\star}}(s')} \\
&\quad + \alpha D_{\mathrm{KL}}\left(\pi(\cdot|s')||\pi^{\star}(\cdot|s')\right) \Bigg] \\
&= \frac{1}{1-\gamma} \sum_{s'} d_s^{\pi}(s') \cdot \Bigg[ \underbrace{\sum_{a'} (\pi^{\star}(a'|s') - \pi(a'|s')) \cdot V^{\pi^{\star}}(s')}_{=0} + \alpha D_{\mathrm{KL}}\left(\pi(\cdot|s')||\pi^{\star}(\cdot|s')\right) \Bigg] \\
&= \frac{1}{1-\gamma} \sum_{s'} \left[ d_s^{\pi}(s') \cdot \alpha D_{\mathrm{KL}}\left(\pi(\cdot|s')||\pi^{\star}(\cdot|s')\right) \right],
\end{aligned}
\tag{70}
$$

taking expectation $s \sim \rho$ yields

$$V^{\pi^{\star}}(\rho) - V^{\pi}(\rho) = \frac{1}{1-\gamma} \sum_{s} \left[ d_{\rho}^{\pi}(s) \cdot \alpha \cdot D_{\mathrm{KL}}\left(\pi(\cdot|s)||\pi^{\star}(\cdot|s)\right) \right], \tag{71}$$

which completes the proof. ∎

## APPENDIX C    SUPPORTING LEMMAS

### C.1    BELLMAN CONSISTENCY EQUATION OF MAXENT RL

**Lemma C.1 (Contraction of Soft Value Iteration)** *From (48) and (6), the soft value iteration operator $\mathcal{T}$ defined as*

$$\mathcal{T}Q(s, a) := r(s, a) + \gamma \alpha \mathbb{E}_{s'} \left[ \log \sum_{a'} \exp\left(Q(s', a')/\alpha\right) \right] \tag{72}$$

*is a contraction.*

**Proof** A similar proof appears in Haarnoja (2018), we provide the proof for completeness. To see (72) is a contraction, for each $(s, a) \in \mathcal{S} \times \mathcal{A}$, we have

$$
\begin{aligned}
\mathcal{T}Q_1(s, a) &= r(s, a) + \gamma\alpha \log \sum_{a'} \exp\left(\frac{Q_1(s, a)}{\alpha}\right) \\
&\leq r(s, a) + \gamma\alpha \log \sum_{a'} \exp\left(\frac{Q_2(s, a) + \|Q_1 - Q_2\|_\infty}{\alpha}\right) \\
&\leq r(s, a) + \gamma\alpha \log \left\{ \exp\left(\frac{\|Q_1 - Q_2\|_\infty}{\alpha}\right) \sum_{a'} \exp\left(\frac{Q_2(s, a)}{\alpha}\right) \right\} \\
&= \gamma \|Q_1 - Q_2\|_\infty + r(s, a) + \gamma\alpha \log \sum_{a'} \exp\left(\frac{Q_2(s, a)}{\alpha}\right) = \gamma \|Q_1 - Q_2\|_\infty + \mathcal{T}Q_2(s, a),
\end{aligned}
\tag{73}
$$

which implies $\mathcal{T}Q_1(s, a) - \mathcal{T}Q_2(s, a) \leq \gamma \|Q_1 - Q_2\|_\infty$. Similarly, we also have $\mathcal{T}Q_2(s, a) - \mathcal{T}Q_1(s, a) \leq \gamma \|Q_1 - Q_2\|_\infty$, hence we conclude that

$$
|Q_1(s, a) - Q_2(s, a)| \leq \gamma \|\mathcal{T}Q_1 - \mathcal{T}Q_2\|_\infty, \ \forall (s, a) \in \mathcal{S} \times \mathcal{A}, \tag{74}
$$

which implies $\|Q_1 - Q_2\|_\infty \leq \gamma \|\mathcal{T}Q_1 - \mathcal{T}Q_2\|_\infty$. Hence $\mathcal{T}$ is a $\gamma$-contraction and the optimal policy $\pi^\star$ of it is unique. ∎

### C.2 Constant Minimum Policy Probability

**Lemma C.2 (Lemma 16 of Mei et al. (2020))** *Using the policy gradient method (Algorithm 3) with an initial distribution $\rho$ such that $\rho(s) > 0, \forall S$, we have*

$$
c := \inf_{t \geq 1} \min_{s, a} \pi_{\theta_t}(a|s) > 0 \tag{75}
$$

*is a constant that does not dependent on $t$.*

**Remark C.3 (State Space Dependency of constant $c$)** *Note that assuming $c$ is independent of $S$ is quite strong and does not generally hold as suggested by Li et al. (2021). Still, if one replace the constant $c$ with other $S$ dependent function $f(S)$, one still can apply a similar proof technique for Theorem 4.1 to show that ROLLIN reduces the iteration complexity, and the final iteration complexity bound in Theorem 4.1 will include an additional $f(S)$.*

## APPENDIX D  SUPPORTING ALGORITHMS

---

**Algorithm 3** PG for $\alpha$-MaxEnt RL (Algorithm 1 in Mei et al. (2020))

---

1: **Input:** $\rho, \theta_0, \eta > 0$.
2: **for** $t = 0, \ldots, T$ **do**
3:     $\theta_{t+1} \leftarrow \theta_t + \eta \cdot \frac{\partial V^{\pi_{\theta_t}}(\rho)}{\partial \theta_t}$
4: **end for**

---

**Algorithm 4** Two-Phase SPG for $\alpha$-MaxEnt RL (Algorithm 5.1 in Ding et al. (2021))

---

1: **Input:** $\rho, \theta_0, \alpha, B_1, B_2, T_1, T, \{\eta_t\}_{t=0}^T$
2: **for** $t = 0, 1, \ldots, T$ **do**
3:     **if** $t \leq T_1$ **then**
4:         $B = B_1$                                                              ▷ Phase 1
5:     **else**
6:         $B = B_2$                                                              ▷ Phase 2
7:     **end if**
8:     Run random horizon SPG with $\rho, \alpha, \theta_t, B, t, \eta_t$         ▷ Algorithm 5
9: **end for**

---

---

**Algorithm 5** Random-horizon SPG for $\alpha$-MaxEnt RL Update (Algorithm 3.2 in Ding et al. (2021))

---

1: **Input:** $\rho, \alpha, \theta_0, B, t, \eta_t$
2: **for** $i = 1, 2, ..., B$ **do**
3:      $s^i_{H_t}, a^i_{H_t} \leftarrow \text{SamSA}(\rho, \theta_t, \gamma)$                          $\triangleright$ Algorithm 6
4:      $\hat{Q}^{\pi_{\theta_t}, i} \leftarrow \text{EstEntQ}(s^i_{H_t}, a^i_{H_t}, \theta_t, \gamma, \alpha)$               $\triangleright$ Algorithm 7
5: **end for**
6: $\theta_{t+1} \leftarrow \theta_t + \frac{\eta_t}{(1-\gamma)B} \sum_{i=1}^{B} \left[ \nabla_\theta \log \pi_{\theta_t}(a^i_{H_t}|s^i_{H_t}) \left( \hat{Q}^{\pi_{\theta_t}, i} - \alpha \log \pi_{\theta_t} \right) (a^i_{H_t}|s^i_{H_t}) \right]$

---

**Remark D.1** *Lemma 3.4 in Ding et al. (2021) implies that the estimator*

$$\frac{1}{(1-\gamma)} \left[ \nabla_\theta \log \pi_{\theta_t}(a^i_{H_t}|s^i_{H_t}) \left( \hat{Q}^{\pi_{\theta_t}, i} - \alpha \log \pi_{\theta_t} \right) (a^i_{H_t}|s^i_{H_t}) \right] \tag{76}$$

*in line 6 of Algorithm 6 is an unbiased estimator of the gradient $\nabla_\theta V^{\pi_\theta}(\rho)$.*

---

**Algorithm 6** SamSA: Sample $s, a$ for SPG (Algorithm 8.1 in Ding et al. (2021))

---

1: **Input:** $\rho, \theta, \gamma$
2: Draw $H \sim \text{Geom}(1-\gamma)$         $\triangleright \text{Geom}(1-\gamma)$ geometric distribution with parameter $1-\gamma$
3: Draw $s_0 \sim \rho, a_0 \sim \pi_\theta(\cdot|s_0)$
4: **for** $h = 1, 2, \ldots, H - 1$ **do**
5:      Draw $s_{h+1} \sim \mathbb{P}(\cdot|s_h, a_h), a_{h+1} \sim \pi_{\theta_t}(\cdot|s_{h+1})$
6: **end for**
7: **Output:** $s_H, a_H$

---

**Algorithm 7** EstEntQ: Unbiased Estimation of MaxEnt Q (Algorithm 8.2 in Ding et al. (2021))

---

1: **Input:** $s, a, \theta, \gamma, \alpha$
2: Initialize $s_0 \leftarrow s, a_0 \leftarrow a, \hat{Q} \leftarrow r(s_0, a_0)$
3: Draw $H \sim \text{Geom}(1-\gamma)$
4: **for** $h = 0, 1, \ldots, H - 1$ **do**
5:      $s_{h+1} \sim \mathbb{P}(\cdot|s_h, a_h), a_{h+1} \sim \pi_\theta(\cdot|s_{h+1})$
6:      $\hat{Q} \leftarrow \hat{Q} + \gamma^{h+1}/2 \left[ r(s_{h+1}, a_{h+1}) - \alpha \log \pi_\theta(a_{h+1}|s_{h+1}) \right]$
7: **end for**
8: **Output:** $\hat{Q}$

---

## APPENDIX E    EXPERIMENTAL DETAILS

We use the SAC implementation from `https://github.com/ikostrikov/jaxrl` (Kostrikov, 2021) for all our experiments in the paper. The exception is the goal reaching task with automated curriculum generation, where we use our customized DDPG implementation that is highly inspired from the same codebase.

### E.1    GOAL REACHING WITH AN ORACLE CURRICULUM

For our `antmaze-umaze` experiments with oracle curriculum, we use a sparse reward function where the reward is $0$ when the distance $D$ between the ant and the goal is greater than $0.5$ and $r = \exp(-5D)$ when the distance is smaller than or equal to $0.5$. The performance threshold is set to be $R = 200$. Exceeding such threshold means that the ant stays on top of the desired location for at least 200 out of 500 steps, where 500 is the maximum episode length of the `antmaze-umaze` environment. We use the average return of the last 10 episodes and compare it to the performance threshold $R$. For both of the SAC agents, we use the same set of hyperparameters shown in Table 3. See Algorithm 8, for a more detailed pseudocode.

| | | |
|---|---|---|
| **Initial Temperature** | | 1.0 |
| **Target Update Rate** | update rate of target networks | 0.005 |
| **Learning Rate** | learning rate for the Adam optimizer | 0.0003 |
| **Discount Factor** | | 0.99 |
| **Batch Size** | | 256 |
| **Warmup Period** | number of steps of initial random exploration (random actions) | 10000 |
| **Network Size** | | $(256, 256)$ |

Table 3 Hyperparameters used for the SAC algorithm (Haarnoja et al., 2018)

---

**Algorithm 8** Practical Implementation of ROLLIN

---

1: **Input:** $\{\omega_k\}_{k=0}^K$: input curriculum, $\rho$: initial state distribution, $R$: near-optimal threshold, $\beta$: roll-in ratio, discount factor $\gamma$.
2: Initialize $\mathcal{D} \leftarrow \emptyset, \mathcal{D}_{\exp} \leftarrow \emptyset, k \leftarrow 0$, and two off-policy RL agents $\pi_{\text{main}}$ and $\pi_{\exp}$.
3: **for** each environment step **do**
4:     **if** episode terminating or beginning of training **then**
5:         **if** average return of the last 10 episodes under context $\omega_k$ is greater than $R$ **then**
6:             $k \leftarrow k + 1, \mathcal{D}_{\exp} \leftarrow \emptyset$
7:             Re-initialize the exploration agent $\pi_{\exp}$
8:         **end if**
9:         Start a new episode under context $\omega_k$ with $s_0 \sim \rho, t \leftarrow 0$
10:         **if** $k > 0$ and with probability of $\beta$ **then**
11:             enable **Rollin** for the current episode.
12:         **else**
13:             disable **Rollin** for the current episode.
14:         **end if**
15:     **end if**
16:     **if Rollin** is enabled for the current episode **then**
17:         **if Rollin** is stopped for the current episode **then**
18:             $a_t \sim \pi_{\exp}(a_t|s_t, \omega_k)$
19:         **else**
20:             $a_t \sim \pi_{\text{main}}(a_t|s_t, \omega_{k-1})$
21:             with probability of $1 - \gamma$, stop **Rollin** for the current episode
22:         **end if**
23:     **else**
24:         $a_t \sim \pi_{\text{main}}(a_t|s_t, \omega_k)$
25:     **end if**
26:     take action $a_t$ in the environment and receives $s_{t+1}$ and $r_t = r_{\omega_k}(s_t, a_t)$
27:     add $(s_t, a_t, s_{t+1}, r_t)$ in replay buffer $\mathcal{D}$
28:     **if Rollin** is disabled for the current episode **then**
29:         update $\pi_{\text{main}}$ using $\mathcal{D}$.
30:     **end if**
31:     **if** $\pi_{\exp}$ was used to produce $a_t$ **then**
32:         add $(s_t, a_t, s_{t+1}, r_t)$ in replay buffer $\mathcal{D}_{\exp}$
33:         update $\pi_{\exp}$ using $\mathcal{D}_{\exp}$.
34:     **end if**
35:     $t \leftarrow t + 1$
36: **end for**
37: **Output:** $\pi_{\text{main}}$

---

### E.2   GOAL REACHING WITH AN AUTOMATICALLY GENERATED CURRICULUM

For our `antmaze-umaze` experiments with automatically generated curriculum, we use a sparse reward function where the reward is $-1$ when the distance between the ant and the goal is greater than $1.0$ (this is different from the oracle curriculum setting where the threshold is $0.5$), and $0$ otherwise. We closely follow the DDPG setup in the MEGA paper (Pitis et al., 2020) and use the `rfaab` relabeling with a ratio of $(0.1, 0.4, 0.1, 0.3, 0.1)$. We also use the go-explore style exploration where the epsilon-greedy ratio is increased from $0.1$ to $0.11$ when the goal is reached (whenever a reward of zero is received). The maximum episode length is 500 and we estimate the success rate of the agent by rolling out the policy 10 times with different pairs of random initial starting locations (a square region centered around $(0.0, 0.0)$ with length $2.0$) and goals (a square region centered around $(0.0, -8.0)$ with length $2.0$). See Figure 5 for a visualization of 8 successful trajectories of this task. We use three-layer network $(512, 512, 512)$ with layer normalization for both the actor and the critic. We also use target $Q$ value clipping such that the target $Q$ value that gets backed up is always within $[-100, 0]$ (the range of the achievable $Q$ value with a discount factor of $0.99$ and binary reward $\{-1., 0.\}$). We did not use a dynamic normalization of the observation (which was used in the original paper). Also, we used tanh to squash the output of the actor to be between $-1$ and $1$ rather than using $L_2$ action

penalty as done in the original paper. We include a detailed description of how ROLLIN is applied on top of MEGA in Algorithm 9 with $N = 10^5$ and the goal proposer from the MEGA paper with the default parameters. See Figure 4 for the results of ROLLIN on top of MEGA over a range of $\beta$ values.

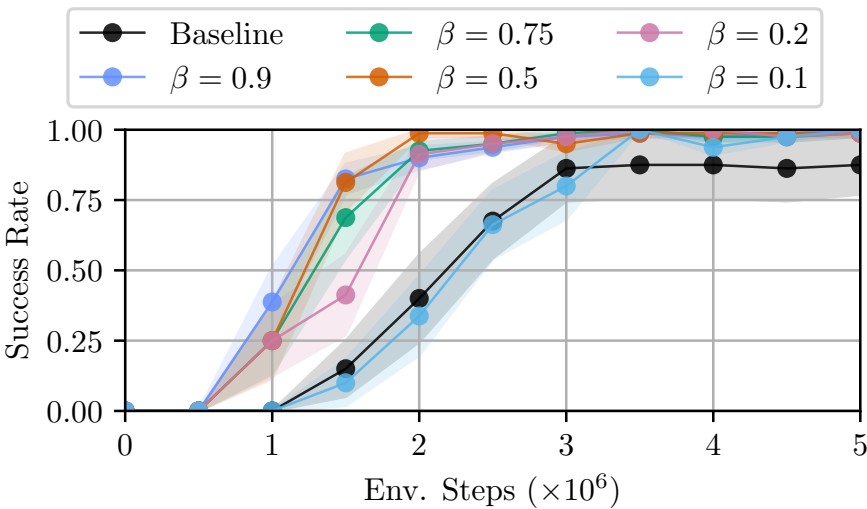

Figure 4 The success rate of goal reaching in `antmaze-umaze` with a curriculum generated by MEGA with $\beta = 0.1, 0.2, 0.5, 0.75, 0.9$. The success rate is reported over 8 independent random seeds. 10 random trials for each seed are used to estimate the success rate.

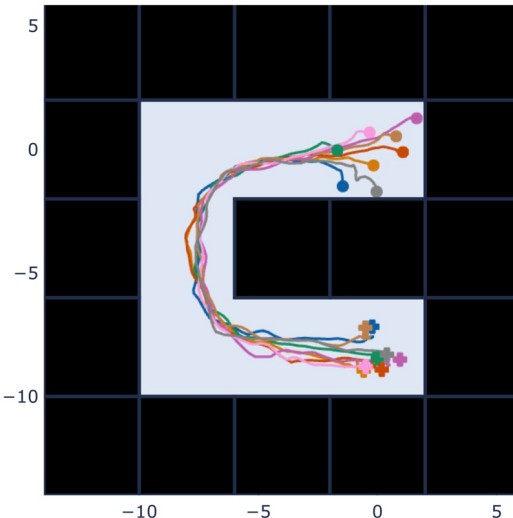

Figure 5 Visualization of the successful trajectories by taking the ant's 2-D location in the maze. The maze walls are colored in black. The circles represent the starting locations of the agent and the crosses with the same color represent their corresponding goal locations.

---

**Algorithm 9** Practical Implementation of ROLLIN when combined with Automated Goal Curriculum Generation

---

1: **Input:** $\rho$: initial state distribution, $\beta$: roll-in ratio, discount factor $\gamma$, goal proposer $\mathcal{G}$, (e.g., MEGA (Pitis et al., 2020)), curriculum step interval $N$,
2: Initialize $\mathcal{D} \leftarrow \emptyset, \mathcal{D}_{\mathrm{exp}} \leftarrow \emptyset, k \leftarrow 0$, and two off-policy RL agents $\pi_{\mathrm{main}}$ and $\pi_{\mathrm{exp}}$.
3: **for** each environment step **do**
4:     **if** episode terminating or beginning of training **then**
5:         **if** more than $N$ steps have been taken at curriculum step $k$ **then**
6:             $k \leftarrow k + 1, \mathcal{D}_{\mathrm{exp}} \leftarrow \emptyset$
7:             Re-initialize the exploration agent $\pi_{\mathrm{exp}}, \pi_{\mathrm{main},k} = \pi_{\mathrm{main},k-1}$
8:         **end if**
9:         Start a new episode with $s_0 \sim \rho, t \leftarrow 0$, get a new goal $g$ proposed by $\mathcal{G}$.
10:         **if** $k > 0$ and with probability of $\beta$ **then**
11:             enable **Rollin** for the current episode.
12:         **else**
13:             disable **Rollin** for the current episode.
14:         **end if**
15:     **end if**
16:     **if Rollin** is enabled for the current episode **then**
17:         **if Rollin** is stopped for the current episode **then**
18:             $a_t \sim \pi_{\mathrm{exp}}(a_t|s_t, g)$
19:         **else**
20:             $a_t \sim \pi_{\mathrm{main},k-1}(a_t|s_t, g)$
21:             with probability of $1 - \gamma$, stop **Rollin** for the current episode
22:         **end if**
23:     **else**
24:         $a_t \sim \pi_{\mathrm{main},k}(a_t|s_t, g)$
25:     **end if**
26:     take action $a_t$ in the environment and receives $s_{t+1}$
27:     add $(s_t, a_t, s_{t+1}, g)$ in replay buffer $\mathcal{D}$
28:     **if Rollin** is disabled for the current episode **then**
29:         update $\pi_{\mathrm{main}}$ using $\mathcal{D}$ with goal relabeling.
30:     **end if**
31:     **if** $\pi_{\mathrm{exp}}$ was used to produce $a_t$ **then**
32:         add $(s_t, a_t, s_{t+1}, g)$ in replay buffer $\mathcal{D}_{\mathrm{exp}}$
33:         update $\pi_{\mathrm{exp}}$ using $\mathcal{D}_{\mathrm{exp}}$ with goal relabeling.
34:     **end if**
35:     $t \leftarrow t + 1$
36: **end for**
37: **Output:** $\pi_{\mathrm{main},k}$

---

### E.3 NON GOAL REACHING

For the non goal reaching tasks in `walker2d`, `hopper`, `humanoid`, and `ant` experiments, the desired $x$-velocity range $[\lambda\kappa, \lambda(\kappa + 0.1))$, the near-optimal threshold $R(\kappa)$, and the `healthy_reward` all depend on the environments. The maximum episode length 1000. Details are provided in Table 4.

| Env. | $\lambda$ | $R(\kappa)$ | healthy_reward | | |
| --- | --- | --- | --- | --- | --- |
| | | | original | high | low |
| walker | 5 | $500 + 4500\kappa$ | 1.0 | 1.5 | 0.5 |
| hopper | 3 | $500 + 4500\kappa$ | 1.0 | 1.5 | 0.5 |
| humanoid | 1 | $2500 + 2500\kappa$ | 5.0 | 7.5 | 2.5 |
| ant | 6 | $500 + 4500\kappa$ | 1.0 | 1.5 | 0.25 |

Table 4 Learning progress $\kappa$, average $x$-velocity, and average return at the 0.75 and 1.0 million environment steps in walker, hopper, humanoid, and ant. The average $x$-velocity and return are estimated using the last 50k time steps. We pick $\beta = 0.1$ for all experiments using ROLLIN, the results of using other $\beta$s can be found in Table 8, 9, 10 in Appendix F.3. The standard error is computed over 8 random seeds.

## APPENDIX F    ADDITIONAL LEARNING CURVES AND TABLES

### F.1    GOAL REACHING

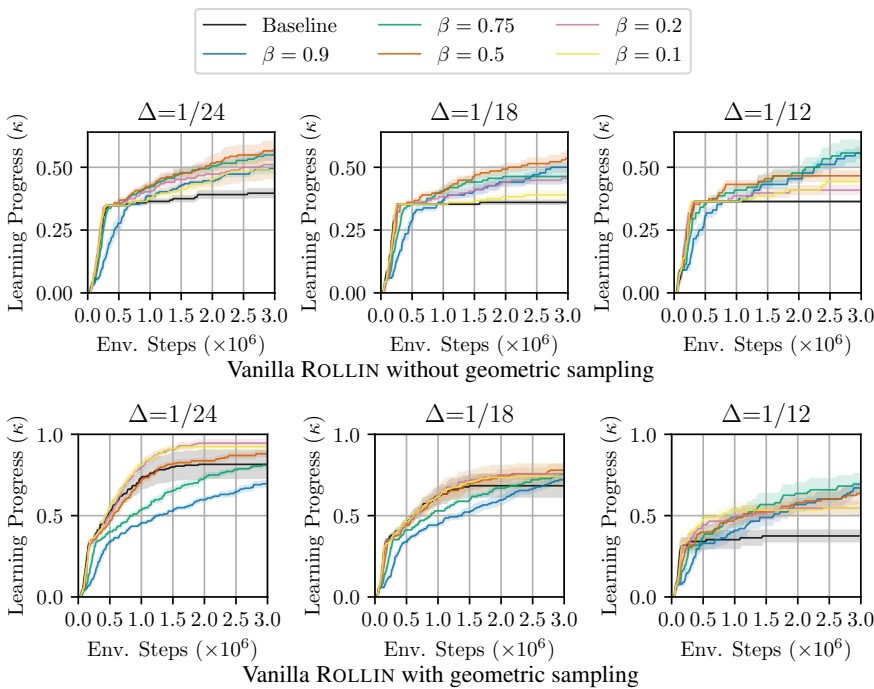

Figure 6 Vanilla Goal reaching. Accelerating learning on antmaze-umaze with ROLLIN on an oracle curriculum in Figure 2. The confidence interval represents the standard error computed over 8 random seeds.

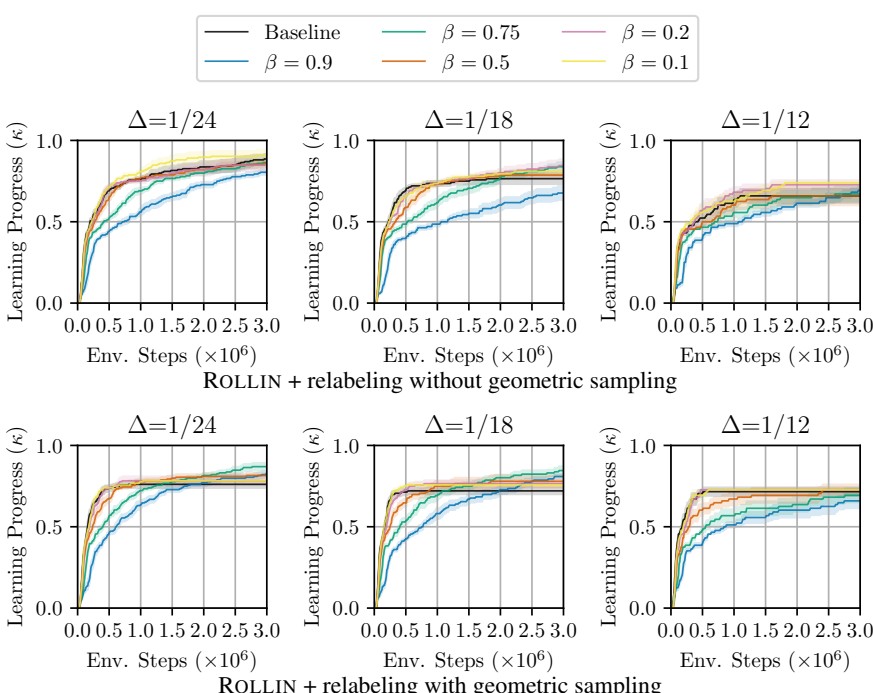

Figure 7 Goal relabeling. Accelerating learning on `antmaze-umaze` with ROLLIN on an oracle curriculum in Figure 2. The confidence interval represents the standard error computed over 8 random seeds.

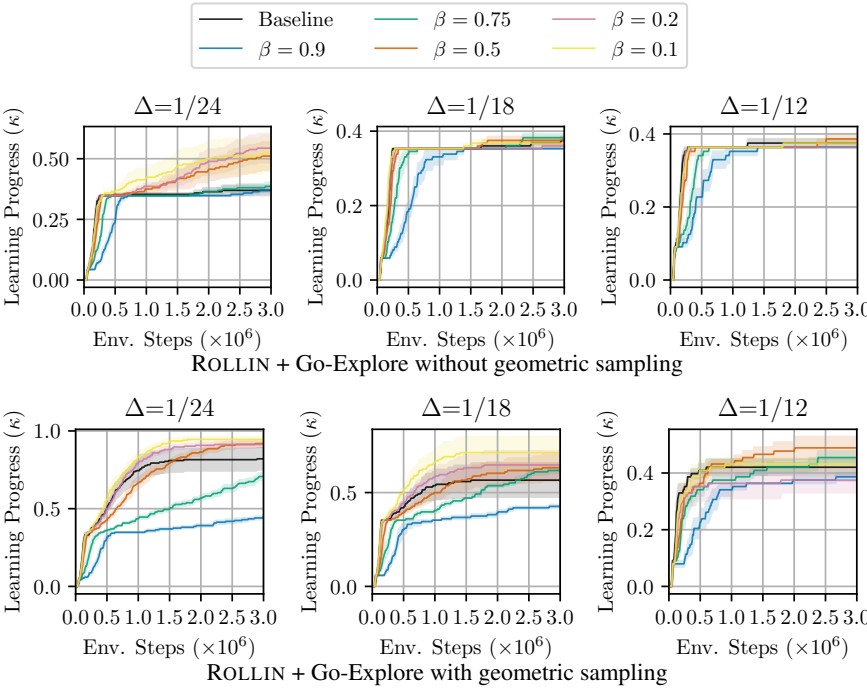

Figure 8 Go-Explore (exploration noise = 0.1). Accelerating learning on `antmaze-umaze` with ROLLIN on an oracle curriculum in Figure 2. The confidence interval represents the standard error computed over 8 random seeds.

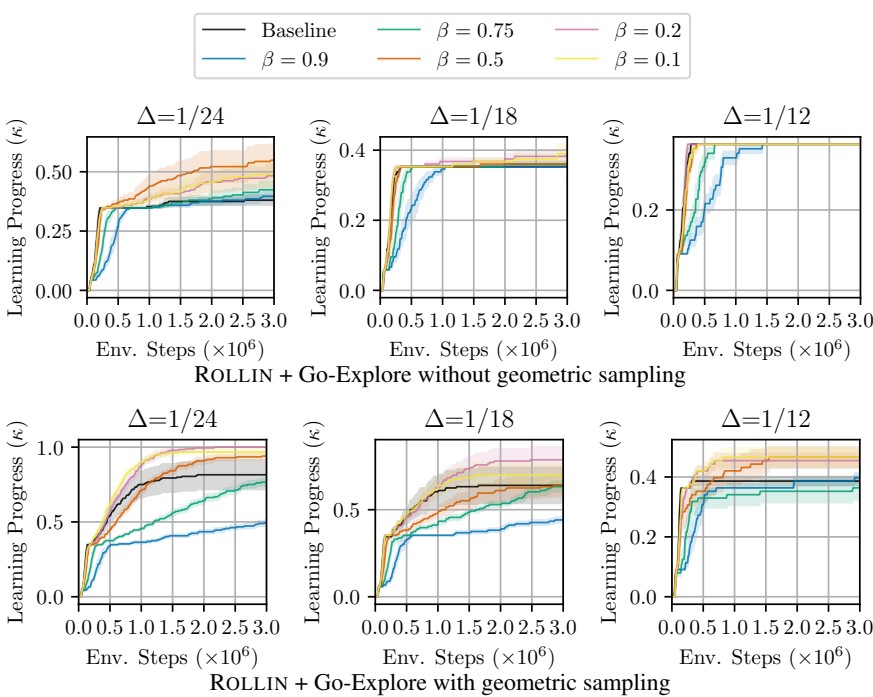

Figure 9 Go-Explore (exploration noise = 0.25). Accelerating learning on `antmaze-umaze` with ROLLIN on an oracle curriculum in Figure 2. The confidence interval represents the standard error computed over 8 random seeds.

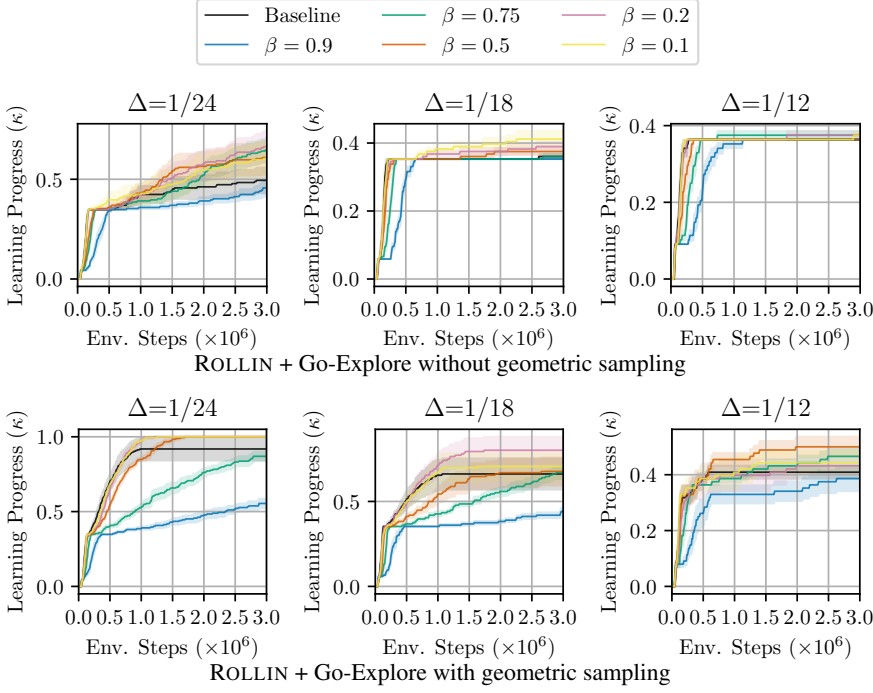

Figure 10 Go-Explore (exploration noise = 0.5). Accelerating learning on `antmaze-umaze` with ROLLIN on an oracle curriculum in Figure 2. The confidence interval represents the standard error computed over 8 random seeds.

| Geo | $\Delta$ | Baseline | $\beta=0.1$ | $\beta=0.2$ | $\beta=0.5$ | $\beta=0.75$ | $\beta=0.9$ |
|---|---|---|---|---|---|---|---|
| ✗ | 1/24 | $0.40\pm0.02$ | $0.49\pm0.04$ | $0.51\pm0.05$ | $\mathbf{0.57\pm0.04}$ | $0.55\pm0.02$ | $0.49\pm0.02$ |
| ✗ | 1/18 | $0.36\pm0.01$ | $0.39\pm0.01$ | $0.46\pm0.01$ | $\mathbf{0.54\pm0.03}$ | $0.46\pm0.03$ | $0.50\pm0.02$ |
| ✗ | 1/12 | $0.36\pm0.00$ | $0.44\pm0.01$ | $0.41\pm0.02$ | $0.47\pm0.02$ | $\mathbf{0.56\pm0.05}$ | $\mathbf{0.56\pm0.02}$ |
| ✓ | 1/24 | $0.82\pm0.08$ | $0.92\pm0.02$ | $\mathbf{0.95\pm0.02}$ | $0.88\pm0.01$ | $0.81\pm0.01$ | $0.70\pm0.02$ |
| ✓ | 1/18 | $0.68\pm0.07$ | $0.74\pm0.07$ | $0.76\pm0.06$ | $\mathbf{0.78\pm0.03}$ | $0.75\pm0.02$ | $0.72\pm0.02$ |
| ✓ | 1/12 | $0.38\pm0.03$ | $0.55\pm0.04$ | $0.55\pm0.04$ | $0.64\pm0.06$ | $\mathbf{0.69\pm0.06}$ | $0.67\pm0.03$ |

Table 5 Vanilla Goal reaching. Learning progress $\kappa$ at 3 million environment steps with varying $\beta$ and curriculum step size $\Delta$ of vanilla goal reaching task. Geo indicates the usage of geometric sampling. Baseline corresponds to $\beta=0$, where no ROLLIN is used. The standard error is computed over 8 random seeds. We highlight the values that are larger than the baseline ($\beta=0$) in purple, and the largest value in **bold font**.

| Geo | $\Delta$ | $\beta=0$ | $\beta=0.1$ | $\beta=0.2$ | $\beta=0.5$ | $\beta=0.75$ | $\beta=0.9$ |
|---|---|---|---|---|---|---|---|
| ✗ | 1/24 | $0.89\pm0.03$ | $\mathbf{0.91\pm0.03}$ | $0.85\pm0.04$ | $0.86\pm0.02$ | $0.86\pm0.02$ | $0.80\pm0.02$ |
| ✗ | 1/18 | $0.76\pm0.03$ | $0.81\pm0.01$ | $\mathbf{0.85\pm0.04}$ | $0.79\pm0.01$ | $0.84\pm0.03$ | $0.68\pm0.04$ |
| ✗ | 1/12 | $0.66\pm0.04$ | $\mathbf{0.74\pm0.01}$ | $0.73\pm0.03$ | $0.66\pm0.06$ | $0.67\pm0.05$ | $0.69\pm0.03$ |
| ✓ | 1/24 | $0.76\pm0.02$ | $0.78\pm0.01$ | $0.78\pm0.03$ | $0.82\pm0.02$ | $\mathbf{0.87\pm0.02}$ | $0.83\pm0.02$ |
| ✓ | 1/18 | $0.72\pm0.02$ | $0.76\pm0.01$ | $0.76\pm0.02$ | $0.78\pm0.04$ | $\mathbf{0.85\pm0.03}$ | $0.81\pm0.01$ |
| ✓ | 1/12 | $0.72\pm0.03$ | $\mathbf{0.73\pm0.00}$ | $\mathbf{0.73\pm0.00}$ | $\mathbf{0.73\pm0.03}$ | $0.69\pm0.04$ | $0.66\pm0.04$ |

Table 6 Goal relabeling. All other settings are the same as Table 5.

| EN | Geo | $\Delta$ | $\beta=0$ | $\beta=0.1$ | $\beta=0.2$ | $\beta=0.5$ | $\beta=0.75$ | $\beta=0.9$ |
|---|---|---|---|---|---|---|---|---|
| 0.1 | ✗ | 1/24 | $0.37\pm0.02$ | $0.52\pm0.07$ | $\mathbf{0.54\pm0.06}$ | $0.51\pm0.06$ | $0.39\pm0.02$ | $0.37\pm0.01$ |
| 0.1 | ✗ | 1/18 | $0.38\pm0.01$ | $0.37\pm0.01$ | $0.36\pm0.01$ | $0.38\pm0.01$ | $0.38\pm0.01$ | $0.35\pm0.00$ |
| 0.1 | ✗ | 1/12 | $0.38\pm0.01$ | $0.38\pm0.01$ | $0.36\pm0.00$ | $\mathbf{0.39\pm0.01}$ | $0.36\pm0.00$ | $0.36\pm0.00$ |
| 0.1 | ✓ | 1/24 | $0.82\pm0.07$ | $\mathbf{0.95\pm0.02}$ | $0.91\pm0.02$ | $0.92\pm0.02$ | $0.71\pm0.02$ | $0.45\pm0.01$ |
| 0.1 | ✓ | 1/18 | $0.57\pm0.09$ | $\mathbf{0.71\pm0.08}$ | $0.65\pm0.07$ | $0.63\pm0.07$ | $0.62\pm0.02$ | $0.43\pm0.01$ |
| 0.1 | ✓ | 1/12 | $0.42\pm0.03$ | $0.43\pm0.02$ | $0.38\pm0.04$ | $\mathbf{0.49\pm0.04}$ | $0.45\pm0.02$ | $0.39\pm0.01$ |
| 0.25 | ✗ | 1/24 | $0.38\pm0.02$ | $0.49\pm0.06$ | $0.48\pm0.05$ | $\mathbf{0.55\pm0.07}$ | $0.43\pm0.04$ | $0.40\pm0.02$ |
| 0.25 | ✗ | 1/18 | $0.35\pm0.00$ | $\mathbf{0.39\pm0.03}$ | $0.39\pm0.02$ | $0.36\pm0.01$ | $0.36\pm0.01$ | $0.35\pm0.00$ |
| 0.25 | ✗ | 1/12 | $0.36\pm0.00$ | $0.36\pm0.00$ | $0.36\pm0.00$ | $0.36\pm0.00$ | $0.36\pm0.00$ | $0.36\pm0.00$ |
| 0.25 | ✓ | 1/24 | $0.82\pm0.10$ | $0.97\pm0.02$ | $\mathbf{1.00\pm0.00}$ | $0.94\pm0.02$ | $0.77\pm0.02$ | $0.49\pm0.02$ |
| 0.25 | ✓ | 1/18 | $0.64\pm0.10$ | $0.70\pm0.07$ | $\mathbf{0.79\pm0.07}$ | $0.64\pm0.06$ | $0.63\pm0.03$ | $0.44\pm0.01$ |
| 0.25 | ✓ | 1/12 | $0.39\pm0.01$ | $\mathbf{0.47\pm0.03}$ | $0.45\pm0.02$ | $0.47\pm0.03$ | $0.36\pm0.04$ | $0.40\pm0.02$ |
| 0.5 | ✗ | 1/24 | $0.49\pm0.06$ | $0.60\pm0.08$ | $\mathbf{0.66\pm0.08}$ | $0.61\pm0.08$ | $0.65\pm0.06$ | $0.46\pm0.04$ |
| 0.5 | ✗ | 1/18 | $0.36\pm0.01$ | $\mathbf{0.41\pm0.02}$ | $0.39\pm0.01$ | $0.38\pm0.01$ | $0.37\pm0.01$ | $0.35\pm0.00$ |
| 0.5 | ✗ | 1/12 | $0.36\pm0.00$ | $\mathbf{0.38\pm0.01}$ | $0.38\pm0.01$ | $0.36\pm0.00$ | $\mathbf{0.38\pm0.01}$ | $0.36\pm0.00$ |
| 0.5 | ✓ | 1/24 | $0.92\pm0.08$ | $\mathbf{1.00\pm0.00}$ | $\mathbf{1.00\pm0.00}$ | $\mathbf{1.00\pm0.00}$ | $0.87\pm0.03$ | $0.55\pm0.03$ |
| 0.5 | ✓ | 1/18 | $0.66\pm0.09$ | $0.71\pm0.08$ | $\mathbf{0.80\pm0.08}$ | $0.68\pm0.08$ | $0.67\pm0.04$ | $0.44\pm0.02$ |
| 0.5 | ✓ | 1/12 | $0.41\pm0.02$ | $0.44\pm0.04$ | $0.43\pm0.03$ | $\mathbf{0.50\pm0.04}$ | $0.47\pm0.03$ | $0.39\pm0.04$ |

Table 7 Go-Explore with different exploration noise. EN represents the multiplier for the Gaussian exploration noise. All other settings are the same as Table 5.

## F.2 GOAL REACHING WITH A LEARNED CURRICULUM

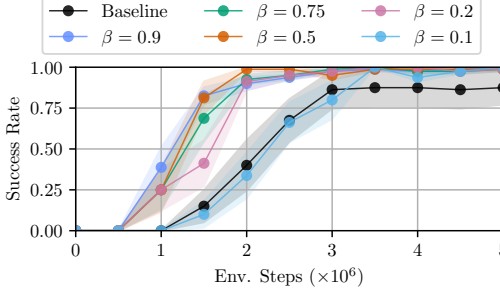

Figure 11 Performance of ROLLIN combined with MEGA (Pitis et al., 2020) on `antmaze-umaze` with different choices of $\beta$.

## F.3 Non Goal Reaching Tasks

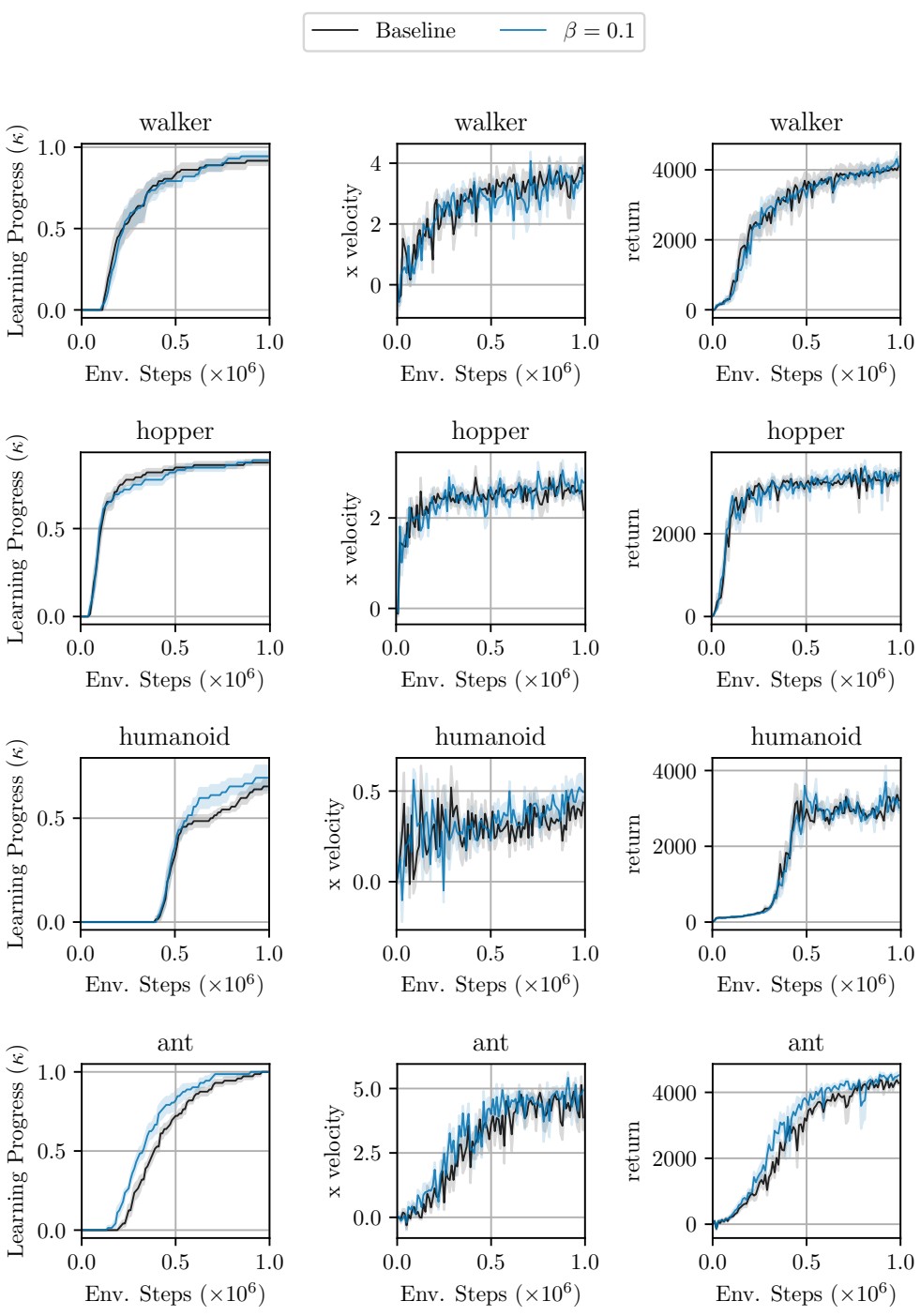

Figure 12 Accelerating learning on several non goal-reaching tasks. The confidence interval represents the standard error computed over 8 random seeds, for $\beta = 0.1$.

| Env. | Step | $\beta = 0$ | $\beta = 0.1$ | $\beta = 0.2$ | $\beta = 0.5$ | $\beta = 0.75$ |
|---|---|---|---|---|---|---|
| walker | 0.75m | $0.89 \pm 0.03$ | $\mathbf{0.90 \pm 0.02}$ | $0.85 \pm 0.03$ | $0.88 \pm 0.04$ | $0.85 \pm 0.04$ |
| | 1m | $0.92 \pm 0.03$ | $\mathbf{0.94 \pm 0.03}$ | $0.90 \pm 0.01$ | $0.92 \pm 0.04$ | $0.92 \pm 0.03$ |
| hopper | 0.75m | $0.86 \pm 0.02$ | $0.85 \pm 0.02$ | $\mathbf{0.88 \pm 0.02}$ | $0.81 \pm 0.03$ | $0.79 \pm 0.01$ |
| | 1m | $0.88 \pm 0.01$ | $\mathbf{0.89 \pm 0.00}$ | $0.89 \pm 0.03$ | $0.82 \pm 0.02$ | $0.81 \pm 0.02$ |
| humanoid | 0.75m | $0.54 \pm 0.01$ | $0.61 \pm 0.05$ | $0.56 \pm 0.03$ | $\mathbf{0.67 \pm 0.03}$ | $0.64 \pm 0.05$ |
| | 1m | $0.67 \pm 0.03$ | $0.69 \pm 0.06$ | $0.62 \pm 0.02$ | $\mathbf{0.76 \pm 0.03}$ | $0.71 \pm 0.06$ |
| ant | 0.75m | $0.93 \pm 0.02$ | $\mathbf{0.99 \pm 0.01}$ | $0.79 \pm 0.09$ | $0.79 \pm 0.06$ | $0.64 \pm 0.07$ |
| | 1m | $1.00 \pm 0.00$ | $1.00 \pm 0.00$ | $0.83 \pm 0.08$ | $0.86 \pm 0.06$ | $0.71 \pm 0.07$ |

Table 8 Learning progress $\kappa$ at 0.75 and 1.0 million environment steps with varying $\beta$ of non goal reaching tasks. Baseline corresponds to $\beta = 0$, where no ROLLIN is used. The standard error is computed over 8 random seeds. We highlight the values that are larger than the baseline ($\beta = 0$) in purple, and the largest value in **bold font**.

| Env. | Step | $\beta = 0$ | $\beta = 0.1$ | $\beta = 0.2$ | $\beta = 0.5$ | $\beta = 0.75$ |
|---|---|---|---|---|---|---|
| walker | 0.75m | $3.20 \pm 0.40$ | $\mathbf{3.48 \pm 0.29}$ | $2.99 \pm 0.26$ | $3.08 \pm 0.30$ | $3.25 \pm 0.38$ |
| | 1m | $3.69 \pm 0.27$ | $3.62 \pm 0.26$ | $3.09 \pm 0.28$ | $3.48 \pm 0.27$ | $3.14 \pm 0.34$ |
| hopper | 0.75m | $2.48 \pm 0.14$ | $2.54 \pm 0.17$ | $2.44 \pm 0.19$ | $2.37 \pm 0.17$ | $\mathbf{2.61 \pm 0.17}$ |
| | 1m | $2.58 \pm 0.16$ | $\mathbf{2.65 \pm 0.15}$ | $2.65 \pm 0.17$ | $2.39 \pm 0.18$ | $2.52 \pm 0.19$ |
| humanoid | 0.75m | $0.32 \pm 0.05$ | $0.39 \pm 0.06$ | $0.34 \pm 0.05$ | $\mathbf{0.41 \pm 0.07}$ | $0.41 \pm 0.09$ |
| | 1m | $0.39 \pm 0.05$ | $0.46 \pm 0.09$ | $0.41 \pm 0.05$ | $0.41 \pm 0.06$ | $\mathbf{0.49 \pm 0.10}$ |
| ant | 0.75m | $4.30 \pm 0.37$ | $\mathbf{4.50 \pm 0.30}$ | $3.55 \pm 0.54$ | $3.47 \pm 0.57$ | $3.21 \pm 0.50$ |
| | 1m | $4.29 \pm 0.51$ | $\mathbf{4.66 \pm 0.30}$ | $3.93 \pm 0.45$ | $3.99 \pm 0.48$ | $3.50 \pm 0.49$ |

Table 9 Average $x$-direction velocity of the last 50k time steps, at 0.75 and 1.0 million environment steps with varying $\beta$ of non goal reaching tasks. Baseline corresponds to $\beta = 0$, where no ROLLIN is used. The standard error is computed over 8 random seeds. We highlight the values that are larger than the baseline ($\beta = 0$) in purple, and the largest value in **bold font**.

| Env. | Step | $\beta = 0$ | $\beta = 0.1$ | $\beta = 0.2$ | $\beta = 0.5$ | $\beta = 0.75$ |
|---|---|---|---|---|---|---|
| walker | 0.75m | $3838.2 \pm 234.0$ | $3831.7 \pm 133.3$ | $3550.3 \pm 115.0$ | $3772.6 \pm 158.6$ | $3667.3 \pm 340.4$ |
| | 1m | $4032.3 \pm 224.3$ | $\mathbf{4128.8 \pm 159.6}$ | $3685.5 \pm 135.6$ | $4028.8 \pm 164.2$ | $3895.4 \pm 265.4$ |
| hopper | 0.75m | $3325.8 \pm 100.6$ | $\mathbf{3339.7 \pm 151.4}$ | $3219.4 \pm 89.4$ | $3008.6 \pm 133.4$ | $3219.2 \pm 142.3$ |
| | 1m | $3386.2 \pm 124.7$ | $\mathbf{3421.9 \pm 109.8}$ | $3262.3 \pm 98.1$ | $3170.7 \pm 180.6$ | $3394.5 \pm 126.5$ |
| humanoid | 0.75m | $3007.5 \pm 176.6$ | $3151.0 \pm 171.6$ | $2848.0 \pm 175.7$ | $3065.2 \pm 226.4$ | $\mathbf{3224.7 \pm 265.2}$ |
| | 1m | $3017.2 \pm 169.0$ | $3173.6 \pm 238.3$ | $2935.8 \pm 181.1$ | $2905.5 \pm 125.9$ | $\mathbf{3290.7 \pm 275.9}$ |
| ant | 0.75m | $3151.0 \pm 224.4$ | $\mathbf{4242.1 \pm 89.5}$ | $3259.8 \pm 434.1$ | $3196.9 \pm 434.6$ | $2739.9 \pm 360.7$ |
| | 1m | $4248.5 \pm 88.6$ | $\mathbf{4473.0 \pm 102.2}$ | $3683.1 \pm 345.0$ | $3708.7 \pm 290.5$ | $3250.1 \pm 316.2$ |

Table 10 Average return of the last 50k time steps, at the 0.75 and 1.0 million environment steps with varying $\beta$ of non goal reaching tasks. Baseline corresponds to $\beta = 0$, where no ROLLIN is used. The standard error is computed over 8 random seeds. We highlight the values that are larger than the baseline ($\beta = 0$) in purple, and the largest value in **bold font**.

## APPENDIX G    ADDITIONAL DISCUSSIONS ON THE RELATED WORK

Liu et al. (2022); Bassich et al. (2020) also consider RL tasks with a curriculum. From a theoretical perspective, we demonstrate the sample complexity gain in the stochastic policy gradient method using non-asymptotic statistical analysis techniques, whereas Liu et al. (2022); Bassich et al. (2020) do not study policy gradient methods.

From an empirical perspective, the non goal-reaching Mujoco locomotion domain is related to the continuous robot evolution model considered by Liu et al. (2022). This prior work uses a context to characterize the transition dynamics of the robot, while we assume the same transition dynamics for all tasks and our context only determines the reward function. The major empirical difference between our work and Liu et al. (2022) lies in the algorithm. ROLLIN facilitates learning by rolling

in a "near-optimal" policy learned from the previous context for learning the next context, which enables the agent to start from an initialization that is "close" to the optimal policy specified by the current context, while Liu et al. (2022) stores all transitions of different robot parameters into one replay buffer, and uses the replay buffer to learn a single policy.

Our goal-reaching tasks in the `antmaze-umaze` environment also appears in Bassich et al. (2020). However, our work studies a problem that is different to Bassich et al. (2020). In particular, Bassich et al. (2020) considers curriculum learning for RL by proposing a progression function that generates the context for each task, while our method tackles the orthogonal problem of using a given curriculum to accelerate RL. Our method is agnostic to where this curriculum comes from, and it could be produced by the designer (Section 6.1) or another algorithm (Section 6.2).

## APPENDIX H    ADDITIONAL DISCUSSIONS ON THE LIMITATIONS

Our theoretical analysis in the tabular case inspires a practical implementation of ROLLIN which demonstrates empirical success in various domains. However, we shall clarify that theoretical analysis in the tabular case is still limited for understanding the empirical success of RL practice, as the state spaces in many RL applications are either enormous or continuous. One potential future direction is to extend the initial distribution update procedure (1) from tabular analysis to feature space analysis.

On the practical side, although ROLLIN generally demonstrates empirical benefits with a choice of $\beta = 0.1$ or $\beta = 0.2$, one still needs to fine tune the $\beta$ for achieving the best performance in general. Hence, another future direction on the empirical side could be implementing the an auto-tuning version of the ROLLIN parameter $\beta$.

