# OpenReview forum: "Understanding the Complexity Gains of Contextual Multi-task RL with Curricula"
_ICLR.cc/2023/Conference — Submitted to ICLR 2023_

### Official Review · Reviewer_haUz · 2022-10-22

**Confidence:** 3
**Correctness:** 3
**Technical Novelty And Significance:** 3
**Empirical Novelty And Significance:** 3
**Recommendation:** 6

**Clarity, Quality, Novelty And Reproducibility:**

### Clarity

The paper is mostly clearly written, but it assumes knowing a lot of context from the reader. Some of the concepts used in the introduction
 get defined only later in the 'Preliminaries' section.

### Quality

I am not a theoretical RL researcher, but the theoretical part looks of high quality to me. My concerns about the assumption on knowing the optimal policy for the first MDP are outlined in the 'weaknesses' section. As for the empirical part, I am concerned with the reported metric, as I also mentioned in the 'weaknesses'

### Novelty

The theoretical part is novel to the best of my knowledge. Empirical part has some missing related works as I pointed out in the 'weaknesses'.

### Reproducibility

The appendix contains detailed pseudocode and the hyper parameters to reproduce the experiments.


**Strength And Weaknesses:**


### Strengths

* The paper proposes a neat way to reduce sample efficiency in the setting providing the theoretical proof (I did not check all the derivations in the appendix).

### Weaknesses

* The paper chooses a non-standard way of evaluating the algorithm, i.e., providing the learning progress value instead of the policy returns. As far as I understand, we have an ordered set of MDPs, and we are eventually interested in solving the target one. I believe, the chosen evaluation protocol favours the proposed method, and I would like to see the comparison: curriculum vs baseline on the target task only.
* The proposed algorithm requires knowing the optimal parameters for the first MDP in the set. If I get it correctly, if we remove this quite limiting assumption, we will get the exponential complexity again, though it will still be reduced by the consequent application of ROLLIN.
* The paper misses important connection to the related work, which is more empirical, but very similar in spirit:
    * REvolveR: Continuous Evolutionary Models for Robot-to-robot Policy Transfer, ICML 2022
    * Curriculum Learning with a Progression Function, Bassich et al.
In addition, I think the paper misleads the reader by its title. Usually, by multitask RL, we mean a single model being able to solve multiple tasks at once, whereas this paper looks at the linear curriculum with a single target task.


**Summary Of The Paper:**

The paper proposes ROLLIN, and algorithm to reduce the complexity of learning an optimal policy in a sequence of contextual MDPs, i.e. a set of MDPs sharing everything but the reward function in this work. The main idea of the paper is that if the two contexts are close enough, we can modify the initial state distribution in a way to reduce the total computational complexity of a policy gradient method.


**Summary Of The Review:**

My current recommendation 'below acceptance' is mostly caused by the empirical evaluation protocol weaknesses. I am ready to increase the score if my concerns from the 'weaknesses' are addressed.

---

> ### Author Response · Authors · 2022-11-13
> **Replies to reviewer haUz (1/2)**
>
> Thank you for your appreciation of the theoretical results and the empirical potential of our work!
>
> **On the empirical side: We have included the requested policy return metrics and the direct target task learning baseline for two additional sets of experiments (where we show ROLLIN is effective on)**. The two additional experiments we added are (1) an extension of our AntMaze result where no oracle curriculum is used, and (2) a set of four challenging MuJoCo locomotion tasks that are modified from the standard Gym tasks with customized oracle reward curricula. In both settings, ROLLIN-augmented learning is effective in accelerating the learning progress, often leading to better final performance compared to learning without ROLLIN.
>
> **On the theoretical side:** To address your concern about the assumption in our theory being too strong, **we have relaxed Assumption 3.2 from “the optimal policy of the first context” to “near-optimal policy of the first context”** to make our results a bit more general.
>
> Please let us know whether the additional experiments and theoretical results have addressed all of your concerns. We would be happy to clarify further if there are other remaining issues!
>
> **Detailed Response:**
> 1. *“The paper chooses a non-standard way of evaluating the algorithm, i.e., providing the learning progress value instead of the policy returns… I believe, the chosen evaluation protocol favors the proposed method, and I would like to see the comparison: curriculum vs baseline on the target task only.”* – Thanks for raising these concerns!
>
> **Baseline on the target task only**:
> We did not show the performance when training on the target task only on the AntMaze environment because the target task (going from the top corner to the bottom corner) is extremely difficult to solve from scratch with sparse reward and usually requires a curriculum of goals to facilitate learning (e.g., see the discussion in paragraph “main results” in Section 4 of [1], which also studies a similar AntMaze environment). We have included a short description in the paper to clarify this. The reason why the target task is hard to learn from scratch is because it is almost impossible to get any reward signal with random exploration in the beginning of the training.
>
> **Policy return metric**:
> We did not include the policy return metric for our AntMaze experiments because most runs did not reach the target task within the 3M environment steps (effectively causing the policy return to be almost all 0 across the board). **We did include the policy return metrics/success in the two additional sets of experiments** that we added for the rebuttal. See below
>
> **MuJoCo Locomotion Tasks with Oracle Reward Curriculum**:
> The first set of experiments includes four MuJoCo locomotion tasks based on the standard Walker2d, Hopper, Humanoid, and Ant Gym tasks with a customized reward curriculum (Section 6.2). Specifically, a higher reward is given at each curriculum step when the desired x-velocity is achieved by the simulated body. As the curriculum progresses, the desired x-velocity increases until the final desired x-velocity is reached. We show that with ROLLIN, the agent is able to progress faster and achieve better final returns, especially on Ant and Humanoid compared to not using ROLLIN (Baseline) or training from scratch (scratch).
>
> **AntMaze with Automatic Goal Generation**:
> In addition to the experiments with oracle rewards, we also experimented with combining ROLLIN with an existing automatic goal generation approach (MEGA [1]) on the same AntMaze task used in our initial submission. In this setting, no oracle curriculum is required. We show that ROLLIN is able to improve MEGA and solves the target tasks much faster (see Section 6.1.2)

---

> > ### Author Response · Authors · 2022-11-13
> > **Replies to reviewer haUz (2/2)**
> >
> > 2. *“The proposed algorithm requires knowing the optimal parameters for the first MDP in the set. If I get it correctly, if we remove this quite limiting assumption, we will get the exponential complexity again, though it will still be reduced by the consequent application of ROLLIN.”* – **We have relaxed the optimal policy assumption in our main theoretical results to require only a near-optimal policy in Assumption 3.2.** Such an assumption is reasonable in many real-world domains, especially for goal-conditioned tasks. For example, we can set the initial context/goal to be close to the starting location of the agent. In this case, making sure that the agent does not move can already give a near-optimal policy.
> >
> > **Detailed Technical Changes**: (1) Instead of assuming the initial parameter $\theta_0^{(0)}$ to be the parameter of $\pi^\star_{\omega_0}$ directly, **we now assume $\theta_0^{(0)}$ to be $\epsilon_0$ optimal**: in Definition 4.2, and $\epsilon_0$ is a constant defined in Theorem A.2 of Appendix A.3. (2) We have updated the proof for the overall complexity for learning the final context $\pi^\star_{\omega_K}$ (Theorem 4.3) using the new $\epsilon_0$-optimality assumption, details can be found in Appendix A.5.
> >
> > 3. *“The paper misses important connection to the related work, which is more empirical, but very similar in spirit” & “In addition, I think the paper misleads the reader by its title”* – Thank you for your suggestion on the writing, we have made the following changes:
> >
> > (1) We **have updated the paragraph “Curriculum learning in reinforcement learning” in Section 2 to include the two papers that the reviewer mentioned [2, 3] and discuss their similarities**.  Empirically, our new experiments in the non-goal environment is similar to [2] – we use a context to specify the desired velocity range of the agent, where we start from a smaller velocity and gradually increase the velocity range; while [2] uses a context to characterize the evolution of the parameters from the source robot to the target robot. Our theoretical formulation is related to [3] – [3] proposes a progression function and a mapping function for curriculum learning in RL, where the progression function generates the curriculum and the mapping function maps the context (or complexity value) to an MDP. This is similar to our setting, where we assume each MDP $\mathcal{M}_{\omega_k}$ is uniquely defined by a context $\omega_k$ in the context space $\Omega$.
> >
> > (2) Sorry for the confusion caused by the title. We have updated it to “Understanding the Complexity Gains of Reformulating Single-Task RL with a Curriculum”.
> >
> > 4. *“Some of the concepts used in the introduction get defined only later in the 'Preliminaries' section.”* – Thank you for your suggestion on the writing, we have updated the introduction accordingly.
> >
> > **References:**
> >
> > [1] Pitis, Silviu, et al. "Maximum entropy gain exploration for long horizon multi-goal reinforcement learning." International Conference on Machine Learning. PMLR, 2020.
> >
> > [2] Liu, Xingyu, Deepak Pathak, and Kris M. Kitani. "REvolveR: Continuous Evolutionary Models for Robot-to-robot Policy Transfer." arXiv preprint arXiv:2202.05244 (2022).
> >
> > [3] Bassich, Andrea, et al. "Curriculum learning with a progression function." arXiv preprint arXiv:2008.00511 (2020).
> >
> > Thank you again for your review and insightful suggestions. We hope that our extra experiments and updated theoretical results address your concern. Please let us know if the updated draft addresses all of your concerns, if so, would you mind updating your recommendation to an *“accept” or higher* based on the theoretical novelty and better empirical results? We would also like to further improve our manuscript or discuss more if you have other suggestions!

---

> > > ### Comment · Reviewer_haUz · 2022-11-14
> > > **response**
> > >
> > > Thanks for adding the related work. However, could you, please, elaborate on the differences between your work and the ones I pointed out in the paper. In other words, what is the exact novelty your work adds on top of what we know after reading Liu/Bassich? What is the difference between your work and the two? Currently, in the paper, you mostly just add the \citep without specifying the details. If you have space limitation, please, add the discussion to the appendix to be moved to the main text when an additional page is allowed.

---

> > > > ### Author Response · Authors · 2022-11-15
> > > > **Followup on the response (2/2)**
> > > >
> > > > Q: *“Thanks for adding the related work. However, could you, please, elaborate on the differences between your work and the ones I pointed out in the paper. In other words, what is the exact novelty your work adds on top of what we know after reading Liu/Bassich? What is the difference between your work and the two? Currently, in the paper, you mostly just add the \citep without specifying the details. If you have space limitation, please, add the discussion to the appendix to be moved to the main text when an additional page is allowed.”*
> > > >
> > > > **Reply**: Thank you for the suggestion, we have added a section in the discussion (Appendix G) to discuss the difference between ROLLIN and [2,3]. We have also attached what we have added below:
> > > >
> > > > ====== Start of Appendix G ======
> > > >
> > > > [2,3] also consider RL tasks with a curriculum. From a theoretical perspective, we demonstrate the sample complexity gain in the stochastic policy gradient method using non-asymptotic statistical analysis techniques, whereas [2,3] do not study policy gradient methods.
> > > >
> > > > From an empirical perspective, the non goal-reaching Mujoco locomotion domain is related to the continuous robot evolution model considered by [2]. This prior work uses a context to characterize the transition dynamics of the robot, while we assume the same transition dynamics for all tasks and our context only determines the reward function. The major empirical difference between our work and [2] lies in the algorithm. ROLLIN facilitates learning by rolling in a "near-optimal" policy learned from the previous context for learning the next context, which enables the agent to start from an initialization that is "close" to the optimal policy specified by the current context, while [2] stores all transitions of different robot parameters into one replay buffer, and uses the replay buffer to learn a single policy.
> > > >
> > > > Our goal-reaching tasks in the antmaze-umaze environment also appear in [3]. However, our work studies a problem that is different to [3]. In particular, [3] considers curriculum learning for RL by proposing a progression function that generates the context for each task, while our method tackles the orthogonal problem of using a given curriculum to accelerate RL. Our method is agnostic to where this curriculum comes from, and it could be produced by the designer (Section 6.1) or another algorithm (Section 6.2).
> > > >
> > > > ====== End of Appendix G ======
> > > >
> > > > Thank you very much for the suggestion, we would be happy to add the updated discussion to the main text if the final version allows an additional page.
> > > >
> > > > [1] Pitis, Silviu, et al. "Maximum entropy gain exploration for long horizon multi-goal reinforcement learning." International Conference on Machine Learning. PMLR, 2020.
> > > >
> > > > [2] Liu, Xingyu, Deepak Pathak, and Kris M. Kitani. "REvolveR: Continuous Evolutionary Models for Robot-to-robot Policy Transfer." arXiv preprint arXiv:2202.05244 (2022).
> > > >
> > > > [3] Bassich, Andrea, et al. "Curriculum learning with a progression function." arXiv preprint arXiv:2008.00511 (2020).
> > > >
> > > > We hope the updated comments and manuscript address all your remaining concerns, please feel free to let us know if you have any other questions or suggestions!

---

> > > > > ### Comment · Reviewer_haUz · 2022-11-15
> > > > > **response**
> > > > >
> > > > > Thanks for the updates.
> > > > >
> > > > > Sorry, I don't think I understand this point:
> > > > >
> > > > > > In particular, [3] considers curriculum learning for RL by proposing a progression function that generates the context for each task, while our method tackles the orthogonal problem of using a given curriculum to accelerate RL. Our method is agnostic to where this curriculum comes from, and it could be produced by the designer (Section 6.1) or another algorithm (Section 6.2).
> > > > >
> > > > > Isn't the point of Bassich to accelerate learning as well (as in any curriculum learning paper)? Progression/mapping functions are just a means to solving the target task (last task of the curriculum. How is that different from ROLLIN?

---

> > > > > > ### Author Response · Authors · 2022-11-15
> > > > > > **Followup on the response**
> > > > > >
> > > > > > Thank you very much for the update.
> > > > > >
> > > > > > A high-level clarification is that: Bassich et al and ROLLIN both aim at facilitating learning, but **ROLLIN facilitates learning in a different way**.
> > > > > >
> > > > > > The two methods are **complementary**, in that they address different aspects of the curriculum learning problem. Rollin does not prescribe a particular curriculum but deals with the question of how each step of the curriculum should be learned efficiently (namely, by rolling in with the policy for the previous step in the curriculum). This approach can be combined with a variety of curriculum choices (e.g., an oracle curriculum discussed in Section 6.1 or a curriculum learned from some other methods discussed in section 6.2). In contrast, the progression function from Bassich et al deals with the question of how to choose the context to train. These are two orthogonal choices for a complete curriculum learning method.
> > > > > >
> > > > > > Perhaps more importantly, our focus is on **deriving provable sample complexity results for Rollin**, while Bassich et al. do not focus on sample complexity analysis. We are not aware of prior work that shows the kinds of sample complexity results for curriculum learning that our paper shows.
> > > > > >
> > > > > > Please let us know whether this clarifies the difference, we are happy to discuss more.

---

> > > > > > > ### Comment · Reviewer_haUz · 2022-11-17
> > > > > > > **response**
> > > > > > >
> > > > > > > Thanks for your response!
> > > > > > >
> > > > > > > I don't have any more questions to the authors and will continue discussing the paper with other reviewers.

---

> > > > > > > > ### Author Response · Authors · 2022-11-17
> > > > > > > > **Thanks!**
> > > > > > > >
> > > > > > > > Thank you again for your time and effort in reviewing our work and the follow-up discussions! If our responses addressed your concerns, we will be grateful if you can re-evaluate our work with an updated score! We will also actively engage in discussions with other reviewers in the future.

---

> > ### Comment · Reviewer_haUz · 2022-11-14
> > **response**
> >
> > Thanks for addressing some of my concerns!
> >
> > > we have relaxed Assumption 3.2 from “the optimal policy of the first context” to “near-optimal policy of the first context” to make our results a bit more general
> >
> > Could you explain, why is this distinction important? From the practical perspective, is there a big difference here?
> >
> > > We did not include the policy return metric for our AntMaze experiments because most runs did not reach the target task within the 3M environment steps
> >
> > Is this the case for some ROLLIN runs as well?
> >
> > > We have included a short description in the paper to clarify this.
> >
> > Could you, please, point me to a specific place in the paper to look at? Or, ideally, highlighting the change with a different colour would be extremely helpful.

---

> > > ### Author Response · Authors · 2022-11-15
> > > **Followup on the response (1/2)**
> > >
> > > We would like to thank the reviewer for the prompt follow-up! To address your follow-up questions/suggestions:
> > >
> > > Q: *“we have relaxed Assumption 3.2 from “the optimal policy of the first context” to “near-optimal policy of the first context” to make our results a bit more general” – Could you explain, why is this distinction important? From the practical perspective, is there a big difference here?*
> > >
> > > **Reply**:  This difference is not major. However, relaxing the requirement to have a perfectly optimal initial policy is more realistic -- it makes it sufficient to have a policy with bounded suboptimality for the initial context (whereas having a perfectly optimal policy might be too strong), which leads to slightly more generalized theoretical results (where we derive a polynomial bound for the final rate of convergence).
> > >
> > > And from a practical perspective, ROLLIN actually works **without** assuming a near-optimal solution as an input. For example, in our updated non-goal reaching Mujoco locomotion tasks, our initialization is actually **not near-optimal** (see our experimental setup in the first paragraph of section 6.3, and appendix E.3). More precisely, to see why the initialization is **not near optimal**, please refer to Figure 12, the learning curves of the non goal reaching tasks in Appendix F.3, where the 4 figures (in walker, hopper, humanoid, ant) on the first column show that the agent requires larger environment steps to learn the first context (especially for the ant and humanoid, where the agent learns the first context at roughly 0.4m and 0.2m environment steps, respectively).
> > >
> > > Q: *“We did not include the policy return metric for our AntMaze experiments because most runs did not reach the target task within the 3M environment steps” – Is this the case for some ROLLIN runs as well?*
> > >
> > > **Reply**: Yes your understanding is correct. To see how far the agent progresses in each different setting, please refer to Table 5-7 and Figures 6-10. When the averaged final learning progress $\kappa$ approaches 1, it means that some of the runs are reaching the final goal (e.g., in Table 7, $\texttt{EN}=0.25, \texttt{Geo}=\checkmark, \Delta=1/24,\beta=0.2$, we have $\kappa = 1$, this results indicate that all runs reach the goal; while for example in the top left entry of Table 7, $0.37\pm0.02$ can be understood as no run reaches the goal).
> > >
> > > For the vanilla goal reaching with an oracle curriculum (Section 6.1), ROLLIN facilitates the learning process by proceeding further; for the curriculum generated by MEGA [1] (Section 6.2, where we provided a success rate), ROLLIN + MEGA improves the success rate of reaching the final goal and facilitate learning progress (see Figure 3).
> > >
> > > Q: *“We have included a short description in the paper to clarify this.” – Could you, please, point me to a specific place in the paper to look at? Or, ideally, highlighting the change with a different colour would be extremely helpful.*
> > >
> > > **Reply**: We have provided the discussion in the last sentence of the last paragraph in Section 6.1 “Goal Reaching with an Oracle Curriculum”, and we have also marked the clarification sentence in blue. For other changes, the reviewer may also use the “compare revisions” function by clicking the "Show Revisions ( https://openreview.net/revisions?id=IW3vvB8uggX )"  link to view the pdf diff generated by openreview.

---

### Official Review · Reviewer_cM8j · 2022-10-25

**Confidence:** 4
**Correctness:** 3
**Technical Novelty And Significance:** 3
**Empirical Novelty And Significance:** 3
**Recommendation:** 6

**Clarity, Quality, Novelty And Reproducibility:**

The presented approach is novel, with results demonstrating improved performance over prior methods.


**Strength And Weaknesses:**

Strengths:
- A novel approach for efficient RL policy learning.
- Algothrim seems sound with theoretical backing.
- Experiments demonstrate that the proposed approach leads to better performance.

Weaknesses
- The experiment section is relatively weak. It is unclear how the proposed approach will scale to other Mujoco robots, such as Swimmer and Humanoid.
- Is the idea of a warm start to policy learning also applicable across different environments? For instance, using parameters from u-maze to some other maze environment could aid in investigating that.


**Summary Of The Paper:**

This paper presents a theoretically-motivated framework that reformulates a single-task RL problem as a multi-task RL problem defined by a curriculum for computationally-efficient policy learning. Their framework and other baseline are demonstrated in a goal-reaching task with a Mujoco dynamical system. The results show improved learning performance compared to traditional methods.

**Summary Of The Review:**

Overall, the presented approach is novel with theoretical support. Experiments demonstrate better performance than the prior methods. However, test cases with different dynamical systems and investigation on the applicability of the proposed approach in speeding up learning across different environments can further strengthen the paper.

---

> ### Author Response · Authors · 2022-11-13
> **Replies to reviewer cM8j**
>
> Thank you for your appreciation of the theoretical results and the empirical potential of our work!
>
> **We have added experiments with more MuJoCo robots* to address your concerns about empirical weakness. In particular, we **construct four additional challenging MuJoCo locomotion tasks** that are modified from the original Gym tasks with customized oracle reward curricula. **We show that ROLLIN-augmented learning is effective in accelerating learning progress**, often leading to better final performance compared to learning without ROLLIN.
>
> **Detailed Response:**
> 1. *“The experiment section is relatively weak. It is unclear how the proposed approach will scale to other Mujoco robots, such as Swimmer and Humanoid.”* – Thank you for your suggestions. We have **added four MuJoCo locomotion tasks based on the standard Gym MuJoCo tasks (Walker, Hopper, Humanoid, and Ant)** with a customized reward curriculum (Section 6.2). In particular, a higher reward is given at each curriculum step when the desired x-velocity is achieved by the simulated body. As the curriculum progresses, the desired x-velocity increases until the final desired x-velocity is reached. **We show that with ROLLIN, the agent is able to progress faster and achieve better final returns**, especially on Ant and Humanoid compared to not using ROLLIN (Baseline) or training from scratch (scratch). See Table 2 for an overview and Appendix F.3 for full results. We would like to emphasize that the paper’s main focus is on the sample complexity analysis, and the empirical results are not intended to be the primary contribution.
>
> 2. *“Is the idea of a warm start to policy learning also applicable across different environments? For instance, using parameters from u-maze to some other maze environment could aid in investigating that”* – Thanks for the suggestion! Warm starting policy learning across different environments with different dynamics/state spaces is definitely an interesting generalization of our framework. However, requires additional assumptions on the MDPs and how the policies transfer across these environments, which we would like to leave for future studies.
>
> Thank you again for your review and insightful suggestions. We hope that our extra experiments and updated theoretical results address your concern. Please let us know if the updated draft addresses all of your concerns, if so, would you mind updating your recommendation to an *“accept” or higher* based on the theoretical novelty and better empirical results? We would also like to further improve our manuscript or discuss more if you have other suggestions!

---

> > ### Author Response · Authors · 2022-11-17
> > **Looking forward to future discussions**
> >
> > Dear reviewer cM8j,
> >
> > Thank you for your time and effort in reviewing our work and the insightful suggestions. We have added several additional experiments based on your suggestions and provided clarifications to your questions. If our response and the updated manuscript have addressed your concerns, we would be grateful if you could reassess our work.
> >
> > Thank you again for your initial endorsement of our work! If you have any additional questions or comments, we would be happy to have further discussions!

---

> > ### Author Response · Authors · 2022-11-18
> > **Follow up**
> >
> > Hi, Reviewer cM8j,
> >
> > We understand that the workload for reviewers is high and we want to appreciate you again for your time.
> >
> > We hope the updated manuscript addresses your previous concerns about our work. We would also like to point out that today is the last day of rebuttal and we would not be able to update the manuscript after today. Please let us know if you have any further questions/concerns about the most updated manuscript so that we can improve it by today.
> >
> > Thank you again!

---

> > ### Comment · Reviewer_cM8j · 2022-11-18
> > **Thank you for your response**
> >
> > Thank you for making revisions. I have the following questions:
> >
> > 1- Is there a standard/generalized way of deciding the value of beta?
> > 2- It seems like in Mujoco environments, the Rollin convergence rate is similar to the baseline. Can the authors provide any reasoning behind it? Is it that Rollin aids learning in some cases and not others?
> > 3- A discussion on the limitations of the proposed work would be great.

---

> > > ### Author Response · Authors · 2022-11-18
> > > **Thanks and followup**
> > >
> > > Thank you for the questions:
> > >
> > > 1. Theoretically, for learning $\omega_k$, the optimal choice of $\beta$ would be $\beta = \frac{1}{\sqrt{C}+1}$, where $C = \frac{\gamma c_1\Delta_\omega^k}{(1-\gamma)\min\mu_k}$. This is obtained by minimizing the RHS of equation (17) in Appendix A.2, where we can choose the best $\beta$ to minimize the upper bound $\frac{1}{\min\mu_k}\frac{\gamma c_1}{(1-\gamma)(1-\beta)}\Delta_\omega^k+\frac{1}{\beta}$ of density mismatch ratio $||d_{\mu_k}^{\pi^\star_{\omega_k}}/\mu_k||_{\infty}$. Empirically we do not have a clear standard on how we should choose the right $\beta$ for achieving the best performance, and we think one future potential direction to improve this work is to develop an auto-tuning procedure for $\beta$, as we have discussed in Section 7 and Appendix H.
> > >
> > > 2. Our conjecture is that ROLLIN has more benefits on "harder" tasks. For example, (1) ROLLIN demonstrates significant improvements in the goal-reaching antmaze tasks; (2) for Mujoco tasks, ROLLIN improves more on "harder tasks" such as humanoid and ant, while only providing marginal improvements in walker and hopper. This is due to the nature of ROLLIN, as ROLLIN works by "rolling in" the policy of a similar task for learning the next task, when the target task can be directly learned from scratch, then ROLLIN only provides limited benefits.
> > >
> > > 3. Thank you very much for raising this point, we have added an additional section in Appendix H to discuss the limitation. And we will like to merge the limitation to the main text if the space permits in the final version.
> > >
> > > Please let us know if there are future questions or suggestions! We would be happy for future discussion or if you can re-evaluate our work based on the updated experiments and the follow-up discussions.
> > >
> > > Thank you!

---

### Official Review · Reviewer_5mDQ · 2022-11-02

**Confidence:** 2
**Clarity, Quality, Novelty And Reproducibility:** See above
**Correctness:** 3
**Technical Novelty And Significance:** 3
**Empirical Novelty And Significance:** Not applicable
**Recommendation:** 5

**Strength And Weaknesses:**

Strengths:
The paper provides novel polynomial time iteration complexity and sample complexity bounds for learning a single-task given a curriculum of contexts.

Weaknesses:
1) Throughout the paper, the authors carry the tone of learning a single-task policy by recasting it as a multi-task problem. (e.g, "In this work, we provide a theoretical framework that reformulates a single-task RL problem as a multi-task RL problem defined by a curriculum.." in the abstract) But, the proposed algorithm "Rollin" assumes that the curriculum (contexts) are given as input.
2) The assumption of having a curriculum that satisfies the assumptions 3.1 and 3.2 is a very strong assumption. In most algorithms involving goal-conditioned RL or curriculum learning, such a curriculum is not provided.
3) They also assume that the optimal policy for the initial context is known.
4) The empirical results are shown on a very simple toy domain.
5) There are multiple grammatical errors / typos throughout the paper.

**Summary Of The Paper:**

Under the assumptions of 1) The reward function is lipschitz continuous w.r.t the context and 2) maximum distance between two contexts, this paper proposes an algorithm "Rollin" to learn an optimal policy for the final context and show that it can be achieved in polynomial iteration complexity instead of exponential. The empirical results were shown on a toy domain.

**Summary Of The Review:**

As listed in the weaknesses, this paper makes very strong assumptions that makes the proposed algorithm impractical to use. The authors should come up with an algorithm to automatically generate the curriculum of contexts. In its current form, I recommend the paper as a weak reject.

---

> ### Author Response · Authors · 2022-11-13
> **Replies to reviewer 5mDQ (1/2)**
>
> Thank you for your appreciation of the theoretical results and the empirical potential of our work!
>
> **On the empirical side:** We have added **two additional experiment settings** to address your concerns about empirical evaluations that are limited domains with the oracle curriculum: 1) an extension of our AntMaze result where no oracle curriculum is used, and 2) a set of four MuJoCo locomotion tasks that are modified from the standard Gym tasks with customized oracle reward curricula. In both settings, **ROLLIN-augmented learning is effective in accelerating the learning progress, often leading to better final performance compared to learning without ROLLIN.**
>
> **On the theoretical side:** To address your concern about the assumption in our theory being too strong, we have **relaxed Assumption 3.2** from “the optimal policy of the first context” to “near-optimal policy of the first context” to make our results a bit more general.
>
> Please let us know whether the additional experiments and theoretical results have addressed your concerns. We would be happy to clarify further if there are other remaining issues!
>
> **Detailed Response:**
>
> 1. *“The empirical results are shown on a very simple toy domain”* – We have **added four MuJoCo locomotion tasks** based on the standard Walker2d, Hopper, Humanoid, and Ant Gym MuJoCo tasks with customized reward curriculum (Section 6.2). In particular, a higher reward is given at each curriculum step when the desired x-velocity is achieved by the simulated body. As the curriculum progresses, the desired x-velocity increases until the final desired x-velocity is reached. **We show that with ROLLIN, the agent is able to progress faster and achieve better final returns, especially on ant and humanoid compared to not using ROLLIN (Baseline) or training from scratch (scratch)**. See Table 2 for an overview and Appendix F.3 for full results. We would emphasize however that the paper’s main focus is on the sample complexity analysis, and the empirical results are not intended to be the primary contribution.
>
> 2. *“The proposed algorithm ‘ROLLIN’ assumes that the curriculum is given as input … In most algorithms involving goal-conditioned RL or curriculum learning, such a curriculum is not provided”* – Our method can accommodate other forms of curriculum that make a sequence of steps from the initial to the final task that satisfy our method’s assumptions, which could include a variety of automated curriculum generation approaches proposed in prior work. For this rebuttal, **we apply ROLLIN on top of a prior automatic goal curriculum generation approach** (MEGA [1]). We experiment on the same AntMaze task where the curriculum is generated automatically by MEGA rather than by hand. See Section 6.1.2 for an overview and Appendix E.2, F.2, for the full procedure and results. **With the addition of ROLLIN, the agent is able to learn faster and achieve a higher final task success rate.**

---

> > ### Author Response · Authors · 2022-11-13
> > **Replies to reviewer 5mDQ (2/2)**
> >
> > 3. *“They also assume that the optimal policy for the initial context is known”* – We have **relaxed the optimal policy assumption in our main theoretical results to require only a near-optimal policy in Assumption 3.2**. Such an assumption is reasonable in many real-world domains, especially for goal-conditioned tasks. For example, we can simply set the initial context/goal to be close to the starting location of the agent. In this case, simply making sure that the agent does not move can already give a near-optimal policy. In terms of the change in our theoretical results in more concrete terms: (1) Instead of assuming the initial parameter $\theta_0^{(0)}$ to be the parameter of $\pi^\star_{\omega_0}$ directly, we now assume $\theta_0^{(0)}$ to be $\epsilon_0$-optimal in Definition 4.2, and $\epsilon_0$ is a constant defined in Theorem A.2 of Appendix A.3.
> > (2) We have updated the proof for the overall complexity for learning the final context $\pi^\star_{\omega_K}$ (Theorem 4.3) using the new $\epsilon_0$-optimality assumption, details can be found in Appendix A.5.
> >
> > 4. *“The assumption of having a curriculum that satisfies assumptions 3.1 and 3.2 is a very strong assumption. In most algorithms involving goal-conditioned RL or curriculum learning, such a curriculum is not provided”* –  We have **updated the paper by adding some discussions after Assumptions 3.1 and 3.2 to clarify the intuition and explain how they help reduce the exponential complexity to a polynomial one**. More concretely: (1) We added a sentence that mentioned several papers (*that we have already cited and discussed in the “contextual MDPs” paragraph of Section 2*) to clarify how the Lipschitz reward assumption appears in prior RL theory literature. (2) We have also updated the paragraph right after Assumption 3.1, 3.2, and section 3.2, by providing more intuition on how they improve the exponential complexity to a polynomial one.
> >
> >
> > **References:**
> >
> > [1] Pitis, Silviu, et al. "Maximum entropy gain exploration for long horizon multi-goal reinforcement learning." International Conference on Machine Learning. PMLR, 2020.
> >
> > Thank you again for your review and insightful suggestions. We hope that our extra experiments and updated theoretical results address your concern. Please let us know if the updated draft addresses all of your concerns, if so, would you mind updating your recommendation to an *“accept” or higher* based on the theoretical novelty and better empirical results? We would also like to further improve our manuscript or discuss more if you have other suggestions!

---

> > ### Author Response · Authors · 2022-11-17
> > **Looking forward to future discussions**
> >
> > Dear reviewer 5mDQ,
> >
> > Thank you for your time and effort in reviewing our work and the insightful suggestions. We have provided detailed clarification to the questions you raised, and updated the manuscript with additional experiments. If our response and the updated manuscript have addressed your concerns, we would be grateful if you could re-evaluate our work.
> >
> > If you have any additional questions or comments, we would be happy to have further discussions!

---

> > ### Author Response · Authors · 2022-11-18
> > **Follow up**
> >
> > Hi, Reviewer 5mDQ,
> >
> > We understand that the workload for reviewers is high and we want to appreciate you again for your time.
> >
> > We hope the updated manuscript addresses your previous concerns about our work. We would also like to point out that today is the last day of rebuttal and we would not be able to update the manuscript after today. Please let us know if you have any further questions/concerns about the most updated manuscript so that we can improve it by today.
> >
> > Thank you again!

---

### Decision · Program_Chairs · 2023-01-20

**Decision:**

Reject

**Justification For Why Not Higher Score:**

The paper indeed provides an interesting approach to learning a single task with a multi-task approach. Yet the theory provided does not seem to be valid as it requires strong assumptions, and the theorems only provide complexity lower bounds (not upper bounds). Some reviewers also believe the contribution (even if correct) is not significant enough.

**Justification For Why Not Lower Score:**

N/A

**Metareview: Summary, Strengths And Weaknesses:**

Summary:

This paper presents a theoretically-motivated framework that reformulates a single-task RL problem as a multi-task RL problem defined by a curriculum for computationally-efficient policy learning. The algorithm has theoretical guarantees under the assumptions of 1) The reward function is lipschitz continuous w.r.t the context and 2) maximum distance between two contexts. The proposed algorithm "Rollin" learn an optimal policy for the final context and the paper shows that it can be achieved in polynomial iteration complexity instead of exponential. The empirical results were shown on some toy environments. During the rebuttal phase, new experiments on mujuco have been added.

Strength:
- A novel approach for efficient RL policy learning with theoretical guarantees.
- Experiments demonstrate that the proposed approach leads to better performance.

Weakness:
- curriculum (contexts) are assumed to be given as input.
- there are multiple grammatical errors/typos throughout the paper.
- the metareview could not verify the theoretical guarantees. Perhaps these are also typos. In Theorem 4.1 and 4.3, why are the complexity bounds presented as $\Omega(\cdot)$? Are these lower bounds? If these are lower bounds, then the theorems do not provide guarantees.



**Summary Of Ac-Reviewer Meeting:**

The reviewer with the most positive review does not have sufficient background in reviewing the theory.